



# Simulating sedimentary burial cycles: Investigating the role of apatite fission track annealing kinetics using synthetic data

Kalin T. McDannell[1] and Dale R. Issler[2]

[1]Department of Earth Sciences, Dartmouth College, Hanover NH, 03755, United States
[2]Geological Survey of Canada, Natural Resources Canada, Calgary AB, T2L 2A7, Canada

*Correspondence to*: Kalin T. McDannell (kalin.t.mcdannell@dartmouth.edu)

**Abstract**

Age dispersion is a common feature of apatite fission track (AFT) and apatite (U–Th)/He (AHe) thermochronological data and it can be attributed to multiple factors. One underappreciated and underreported cause for dispersion is variability in apatite composition and its influence on thermal annealing of fission tracks. Here we investigate, using synthetic data, how multikinetic AFT annealing behaviour (defined using the $r_{mr0}$ parameter) can be exploited to recover more accurate, higher resolution thermal histories than are possible using conventional interpretation and modelling approaches. Our forward model simulation spans a 2 Gyr time interval with two separate heating and cooling cycles and generates synthetic AFT and AHe data for three different apatite populations with significantly different annealing kinetics. The synthetic data are used as input for inverse modelling (Bayesian QTQt model) that attempts to recover thermal history information under various scenarios. Results show that essential features of the dual peak thermal history are captured using the multikinetic AFT data alone, with or without imposed constraints. Best results are achieved when the multikinetic AFT data are combined with the AHe data (using varying $r_{mr0}$ values from the AFT data for the He radiation damage model) and constraints are included. In contrast, a more conventional monokinetic interpretation that ignores multikinetic AFT behaviour yields incorrect thermal solutions that fail to adequately reproduce all the data. The AFT data are reproduced well but the AHe data are not. Under these conditions, incorporation of constraints can be very misleading and fail to improve model results. In general, a close fit between observed and modelled parameters is no guarantee of a robust thermal-history solution if data are incorrectly interpreted. For the case of overdispersed AFT data, it is strongly recommended that elemental data be acquired to investigate if multikinetic annealing is the cause of the age scatter. A future companion paper will explore multikinetic AFT methodology and application to detrital apatite samples from Yukon, Canada.

## 1. Introduction

Studies focusing on upper crustal tectonics, landscape evolution, and sedimentary basin analysis often rely on apatite fission track (AFT) and apatite (U–Th)/He (AHe) low-temperature thermochronology to decipher spatial patterns of exhumation and burial through time (e.g., Ehlers and Farley, 2003; House et al., 1998; Naeser et al., 1989; van der Beek et al., 1995; Zeitler et al., 1982). These low-temperature techniques typically produce internally consistent results in rapidly cooled, actively eroding mountain belts (e.g., Glotzbach et al., 2011), however,



thermochronometric harmony commonly breaks down in slowly cooled settings. There are gaps in our knowledge of
how fission tracks anneal in apatite (e.g., Ketcham, 2019), how $^4$He diffusion occurs over geologic time (e.g.,
McDannell et al., 2018), and if the mechanisms controlling these processes are fundamentally different, linked, or
interact in complex and unforeseen ways. Poorly understood compound variables, both geological and analytical,
sometimes yield apatite thermochronology data that are not straightforward to interpret. For example, AFT < AHe
"age inversion" (e.g., Farley et al., 1996; Fitzgerald et al., 2006; Flowers and Kelley, 2011) is often encountered in
continental interiors and has been attributed to the effects of slow cooling and accumulated radiation damage on He
diffusion (e.g., Green et al., 2006). High age dispersion in AFT data is also seen in slowly cooled, ancient terranes
(McDannell et al., 2019a), suggesting there are unexplained complexities present in both systems.

The canonical temperature sensitivity for AFT dating is ~60–125 °C (Gleadow and Duddy, 1981) and ~45–75 °C for
AHe dating (Wolf et al., 1998). However, temperature sensitivity varies as a function of multiple factors such as
apatite chemistry (Barbarand et al., 2003; Carlson, 1990; Crowley et al., 1990; Green et al., 1985, 1986; Ravenhurst
et al., 1993) and cooling rate for AFT, and radiation damage accumulation, grain size, parent nuclide zoning, and
chemistry for AHe (e.g., Djimbi et al., 2015; Farley, 2000; Gautheron et al., 2013; Gautheron et al., 2009; Recanati
et al., 2017; Shuster et al., 2006). Radiation damage may also play a role in modifying apatite fission track annealing
kinetics from old rocks (e.g., Carpéna et al., 1988; Hendriks and Redfield, 2005), or at least cause reduced thermal
annealing resistance (McDannell et al., 2019a). This is a debated issue (Kohn et al., 2009) requiring further scrutiny
and experimental work to verify empirical relationships (e.g., Carpéna and Lacout, 2010). However, observations of
AHe date–U and date-elemental trends by Recanati et al. (2017) and joint AFT–AHe date–U trends by McDannell et
al. (2019a) imply a complex relationship between α-radiation damage and apatite chemistry, where dates increase
and then decrease as a function of the estimated damage accumulated, similar to observations with zircon
(Guenthner et al., 2013). This suggests a change in both helium and fission-track retention at high radiation damage
levels and warrants a closer inspection of apatite chemistry, radiation damage, and track annealing for applications
in thermal history analysis.

We recognize chemical composition has an effect on both AFT and AHe dates, but careful investigation of this
property for both chronometers remains problematic. The main factors preventing this are practical in nature, in that
most AFT studies utilize compositional proxies due to ease of measurement (i.e., Dpar = mean etch figure width
parallel to $c$-axis; Burtner et al., 1994; Donelick, 1993) and neglect elemental data to fully characterize samples.
Likewise, the bulk AHe method is a destructive technique that precludes single-grain elemental characterization.
The overwhelming majority of published studies featuring age inversion present AFT and AHe data from different
grains, making direct comparisons between individual apatites challenging (Danišík, 2019). There is also the
impractical comparison or statistical problem of likening AFT central ages to mean or single-grain AHe dates. The
central age for AFT data is utilized to provide an approximate geometric mean age for a population of grain ages in
the case of excess age dispersion (Galbraith and Laslett, 1993). Therefore, if an AFT sample fails the $\chi^2$ test and
contains discrete age components or a continuous mixture of ages (Galbraith and Green, 1990; Galbraith and Laslett,





1993), then the meaning of the central age is somewhat misleading for comparative purposes. The same applies to
averaged AHe dates if accumulated radiation damage varies between grains.

In the overall context of age scatter, another equally viable possibility is that overdispersed or inverted dates for the
AFT and AHe thermochronometers occur as a result of variable intrasample retentivity (i.e., resistance to track
annealing and diffusive He loss) for both systems due to the effects of apatite chemical composition. In the simple
case one can imagine two apatite grains, a fluorapatite and the other a more retentive chlorapatite, where the former
is dated by FT and the latter is dated by (U–Th)/He. Assuming a slow-cooling history, the grain dated by AFT may
yield a date that is younger than the grain dated by the (U–Th)/He method solely due to compositional differences.
Here, we present simple examples demonstrating these effects using synthetic AFT and AHe data derived from
forward models utilizing the $r_{mr0}$ kinetic parameter based on apatite composition (Carlson et al., 1999; Ketcham et
al., 1999). The synthetic data are exaggerated, implementing extreme endmember kinetics that are rare, but not
unheard of, in natural crystalline basement samples and more commonly encountered in detrital samples. This was
done to illustrate that multikinetic AFT samples provide an expanded range in thermal sensitivity and that AFT data
may be misrepresented (under the assumption that the central age is wholly descriptive of a sample) if potential
kinetic sub-populations governed by composition are not accounted for during data interpretation or kinetic proxy
data are imprecise (i.e., $D_{par}$; Issler et al., 2018; Schneider and Issler, 2019). Analogously, in the absence of
retentivity information for the AHe system, using a default "fluorapatite" value may completely misrepresent a
sample by introducing modelling artifacts that distort time-temperature (t–T) solutions, or even prevent viable t–T
paths from being found during thermal history analysis. These exercises were performed assuming that we knew the
true thermal history, which is almost always not the case, and they are meant to encourage users of
thermochronology data to more thoroughly interpret data and explore kinetic models before undertaking thermal
history simulations. The results in this paper give us confidence in our treatment of real data and support the idea
that the multikinetic AFT method yields higher resolution thermal histories than the conventional method. In a
future companion paper, we will specifically discuss elemental data collection, multikinetic workflow and
interpretation schemes, and thermal history analysis of natural detrital samples from Yukon, Canada.
**2. Apatite chemistry, track annealing, and the experimentally derived $r_{mr0}$ parameter**
The empirical $r_{mr0}$ kinetic parameter was derived by characterizing track annealing with respect to chemical
composition (Carlson et al., 1999) to produce a multikinetic annealing model (Ketcham et al., 1999). Later work
updated the equation and annealing data fits (Ketcham et al., 2007) by combining the dataset of Barbarand et al.
(2003) with the 1999 dataset. However, the later reformulation of $r_{mr0}$ is different due to the dominant influence of
Cl and OH (and generally lower cation concentrations) in the 2003 dataset, which considerably changes the fitting
parameterization. Although the $r_{mr0}$ kinetic model shows you can reconcile the experimental annealing data with
apatite composition, this does not necessarily mean that more data equates to a better calibration. More data changes
the calibration, but "improvement" depends on whether the calibration data are representative of the natural range of
apatite compositions or are skewed to a particular composition. The Ketcham et al. (2007) model still suffers from





an uneven distribution of data and includes a subset of possible compositional ranges that cause the revised equation
to narrow the range of $r_{mr0}$ slightly from the original model. In our view, the 2007 multikinetic model is no better or
worse than the original model, however the 1999 model is less dominated by chlorapatite compositions, which aids
in clearer multikinetic interpretation (i.e., less kinetic population overlap) for natural AFT samples. It is a reasonable
assumption that the same annealing mechanism and therefore the same kinetic formulation applies to all apatite
varieties. It is primarily our lack of knowledge regarding composition and the relation with annealing, not
necessarily erroneous models that are the main issue for kinetic model calibration. Nevertheless, the utility of the
$r_{mr0}$ function remains for explaining overall apatite annealing–compositional trends. Here we review the $r_{mr0}$
parameter in the context of the original Carlson et al. (1999) expression. The reader is referred to the original papers
or Ketcham (2019) for a comprehensive discussion of $r_{mr0}$.

The $r_{mr0}$ value comes from a simple normalization function that relates one apatite to another for the purpose of
comparing annealing behaviour at laboratory timescales, using the equation:

$$r_{lr} = \left(\frac{r_{mr} - r_{mr0}}{1 - r_{mr0}}\right)^{k}$$    (1)

Where $r_{lr}$ and $r_{mr}$ are the reduced lengths of the apatite that are less resistant and more resistant to annealing,
respectively and $r_{mr0}$ and $k$ are fitted parameters. Specifically, $r_{mr0}$ is the reduced fission-track length of the more
resistant apatite at the point in time and temperature where the less resistant apatite is totally annealed, allowing a
direct comparison between any two apatites (Ketcham et al., 1999). Ketcham et al. (1999) used B2 apatite from
Bamble, Norway (highly enriched in Cl and OH) as the reference datum for $r_{mr0}$, since this was the apatite most
resistant to annealing in the Carlson et al. (1999) experiments. Therefore, $r_{mr0}$ values approaching one, signify lower
retentivity, whereas those approaching zero are more retentive, with common fluorapatite defined by a $r_{mr0}$ value of
0.84. Individual $r_{mr0}$ fits of apatite pairs revealed overall good agreement between measured and predicted mean
(and c-axis projected) lengths, however the simultaneous fits to the entire apatite dataset were lower quality. The
poorer fit was perhaps due to subtle differences in etching/annealing conditions (i.e., temperature control), the
simplification that $r_{mr0} + k \approx 1$, or insufficient compositional diversity and/or elemental data. For example, Si was
not accounted for in the Carlson et al. (1999) and Ketcham et al. (1999) studies but a subsequent study by Tello et al.
(2006) found that Itambé apatite was more resistant to annealing than the Durango apatite laboratory age standard
and is nearly 13x richer in Si (4.15 wt.% Si; simultaneous 1999 fit $r_{mr0}$ = 0.819 excluding Si). Comparing Itambé to
Durango implies higher retentivity for the former, yet the difference in $r_{mr0}$ between Itambé and Durango is very
small using the 1999 $r_{mr0}$ equation (0.31 wt. % Si; simultaneous 1999 fit $r_{mr0}$ = 0.827). The $r_{mr0}$ value for Itambé
calculated using the Ketcham et al. (2007) equation is 0.785; suggesting track retentivity is greater than Durango,
although the 2007 equation is biased towards more retentive apatite. These differences are just one example
indicating further annealing studies are required to account for unusual elemental substitutions that nonlinearly
influence annealing behaviour at the cation sites in apatite (Barbarand et al., 2003; Carlson et al., 1999; Ketcham et
al., 2007).






We utilize the relationship established between $r_{mr0}$ and measured Cl to calculate an "effective Cl" (eCl) value in
atom per formula unit (apfu) from collected electron microprobe data (see McDannell et al., 2019b for further
explanation). Effective Cl is the Cl concentration required to yield an equivalent $r_{mr0}$ value for the Ketcham et al.
(1999) annealing model based on the published correlation between Cl and $r_{mr0}$ in Carlson et al. (1999). The eCl
value (e.g., Issler et al., 2018; McDannell et al., 2019b) is used to transform the nonlinear $r_{mr0}$ parameter to a linear
form for data interpretation using the equation (given in figure 7 of Ketcham et al., 1999):

$$r_{mr0} = 1 - \exp\left[2.107(1 - Cl) - 1.834\right] \qquad (2)$$

In addition, the Ketcham et al. (1999) expression relating $r_{mr0}$ to $D_{par}$ is:

$$r_{mr0} = 1 - \exp\left[0.647(Dpar - 1.75) - 1.834\right] \qquad (3)$$

The constants in these equations changed slightly in the Ketcham et al. (2007) multikinetic model revision but
remain similar to the original calculations. Equations (2) and (3) allow the transformation between measured kinetic
parameters (i.e., $D_{par}$ and Cl) to $r_{mr0}$ and vice versa. For example, the $r_{mr0}$ value from the Ketcham et al. (2007) model
is 0.83 for fluorapatite, which translates to an eCl value of ~0.03 apfu and an eDpar of ~1.85 μm.

Fission-track kinetics have also been used to describe changes in $^4$He diffusivity in the apatite (U–Th)/He system
(e.g. Flowers et al., 2009). The development of a model to explain radiation damage effects on He diffusivity
(Shuster and Farley, 2009; Shuster et al., 2006) resulted in the radiation damage accumulation and annealing model
(RDAAM) by using fission-track annealing kinetics of Ketcham et al. (2007) as a proxy for α-damage or bulk
radiation damage annealing (Flowers et al., 2009). The fundamental assumption being that α-damage and fission-
track damage anneal at the same rate, enabling the use of the $r_{mr0}$ parameter in the RDAAM, set to typical
fluorapatite kinetics ($r_{mr0}$ = 0.83). This allows a comparison between fission track and He data within the same
kinetic framework. However, there is an apparent divergence in damage kinetics, and we now understand that
assuming similar annealing kinetics is an oversimplification and probably incorrect, especially when rocks reside for
long intervals at low temperatures <100°C and at high levels of accumulated radiation damage (Fox and Shuster,
2014; Gautheron et al., 2013; Ketcham, 2019; Ketcham et al., 2017; McDannell et al., 2019a; Recanati et al., 2017;
Willett et al., 2017). However, for common low-damage (i.e., low U) apatite and certain thermal histories, the
kinetics remain valid to first order. We use the $r_{mr0}$ parameter to examine the relationship between apatite
composition and track retentivity (and He diffusivity) and how accounting for or overlooking these associations
influence data interpretation, and ultimately, thermal history modelling results.




**3. Forward and inverse modelling of multikinetic synthetic data**

**3.1 Forward modelled synthetic AFT and AHe data from a predetermined thermal history**

Synthetic AFT data were generated from forward modelling a two-pulse heating history over 2000 Myr using the QTQt software v. 5.7.3 (Gallagher, 2012) implementing Ketcham et al. (1999) annealing kinetics (fig. 1), with one maximum heating event occurring at 1000 Ma (110°C) and the other at 300 Ma (60°C). AFT ages and track length data (fig. 2) were randomly predicted for three kinetic populations as external detector method (EDM) data in QTQt. We specified three AFT kinetic populations of 10 age grains each, increasing in retentivity with $r_{mr0}$ values of 0.882 (eCl = -0.144 apfu), 0.820 (eCl = 0.057 apfu), and 0.263 (eCl = 0.726 apfu) using individual-fit $c$-axis projected length kinetic data for distinct apatites from Ketcham et al. (1999). Population one is set to the Holly Springs (Georgia, USA) hydroxyapatite $r_{mr0}$ that typifies the lowest calculated retentivity in the Carlson et al. (1999) dataset, population two uses Durango apatite kinetics (laboratory age standard), whereas population three is set to Tioga (Pennsylvania, USA) Fe-Cl apatite, which is characterized by high retentivity and is an outlier of the Carlson et al. $r_{mr0}$-fitting dataset. The specified thermal history produced three AFT model ages of 670 Ma, 843 Ma, and 1602 Ma (fig. 2). Seventy-five tracks were generated for each kinetic population with mean $c$-axis projected track lengths (MTL) of 13.32 ± 1.33 μm (1σ), 14.24 ± 1.42 μm, and 14.65 ± 1.47 μm, respectively. The initial (pre-annealed) track lengths ($l_{oc}$) for each kinetic population were calculated as 16.17 μm, 16.40 μm, and 17.16 μm with increasing retentivity and were estimated from the equivalent $D_{par}$ calculated from the indicated $r_{mr0}$ value for each kinetic population (equation 3 above) using the $l_{oc}$–$D_{par}$ relation from Carlson et al. (1999). Three AHe dates were also forward modelled using the radiation damage accumulation and annealing model (RDAAM) of Flowers et al. (2009), which implements the Ketcham et al. (2007) kinetics for radiation damage annealing. We applied Holly Springs, typical endmember fluorapatite ($r_{mr0}$ = 0.83 and the RDAAM default), and Tioga apatite $r_{mr0}$ values to AHe grains, all with spherical grain radii of 50 μm and 25 ppm U (Th and Sm discounted for simplicity). The uncorrected AHe dates (α ejection-corrected date in brackets) were 585 Ma [813 Ma], 610 Ma [848 Ma], and 819 Ma [1139 Ma] predicted using the same t–T history (fig. 1) as the AFT data.

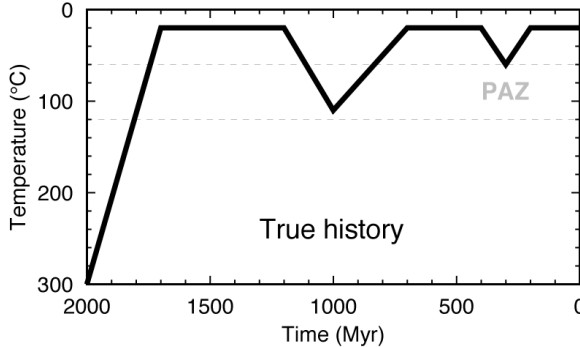

**Figure 1:** Thermal history used to predict synthetic AFT and AHe data. This t–T path is referred to as the "true" thermal history throughout this paper. The predicted synthetic data were then used as input for QTQt to recover the thermal history through inverse modelling. PAZ = partial annealing zone for fission tracks.



### 3.2 Methods for inverting AFT and AHe synthetic data for thermal history

We attempted to recover the true thermal history used to predict the synthetic data from Sect. 3.1 using the QTQt software. These exercises imitate real thermal history investigation in the context of incomplete geologic knowledge, complex or imperfect datasets, and judgement calls that are typically made by researchers implementing thermochronology data and performing modelling to infer quantitative information about geologic processes. We also explore the effects of kinetic assumptions for AHe dates or the consequences of neglecting the identification of multikinetic populations during AFT modelling. An important point is that QTQt will generate thermal histories regardless of feasibility, and it is up to the user to understand the ramifications of this and make sensible decisions about modelling input and output (Gallagher and Ketcham, 2018; Vermeesch and Tian, 2014). We used QTQt because it is sensitive to the number and quality of data during history inference (i.e., notionally improving model results with additional, high quality data) and specifically because it *will* generate model histories regardless of the physical or geologic plausibility for a history simulation — this was done to explore the possible effects of improper data treatment or misinterpretation.

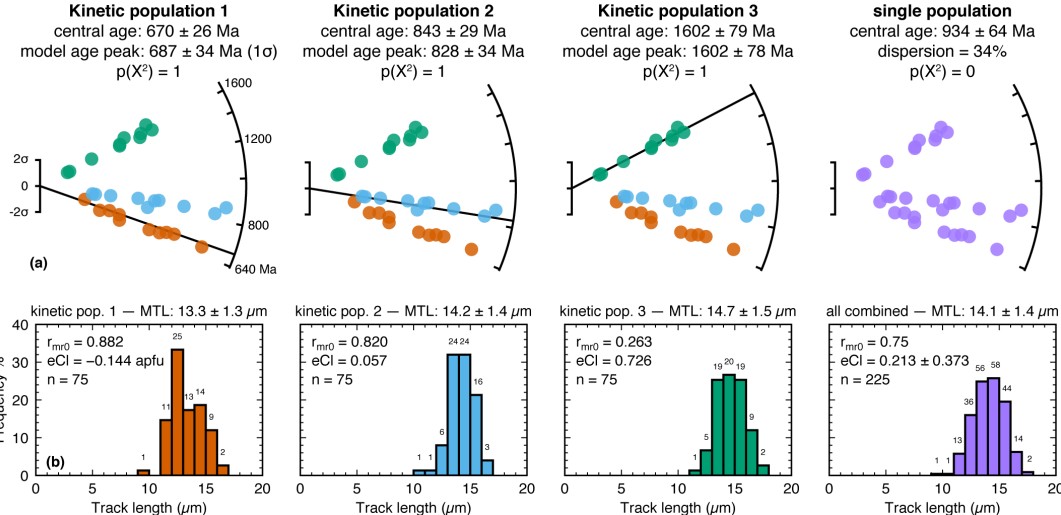

**Figure 2:** Predicted synthetic AFT data from the thermal history in figure 1. Multikinetic age populations were individually predicted using distinct $r_{mr0}$ kinetics shown in (B) panels (discussed in the text). These data were then input in QTQt and inverted in an attempt to recover the true thermal history in figure 1 (see fig. 3). **(A)** Central age and $1\sigma$ errors are indicated for each kinetic population. Kinetic populations one, two, and three are displayed as arms on their respective radial plots, with individual AFT ages closer to the origin being less precise. The last radial plot shows all thirty individual grains and demonstrates that when taken together, the combined sample fails the $\chi^2$ test ($p < 0.05$) for homogeneity (i.e., that all grains belong to a single underlying age population) suggesting multiple age populations. This is the scenario most researchers would start with before evaluating the sample for potential multikinetic behaviour. Mixture modelling was subsequently performed on the combined sample and the model age peaks that were picked seamlessly align with the individual kinetic population central ages. This aligns with how populations would be defined and compared with the elemental chemistry for individual age grains during multikinetic interpretation. **(B)** The predicted track length distributions for each kinetic population from the thermal history in Figure 1 using the specified kinetic parameter value. The last panel on the right combines all tracks from each kinetic population. Numbers on the histogram are the number of tracks in each μm bin. Abbreviations: eCl = effective Cl; MTL = mean track length.

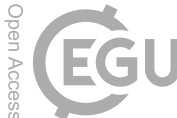

The $r_{mr0}$ values for AFT and AHe data were held fixed for simulations and an appropriate level of noise was added
to the synthetic dataset by adding age scatter to AFT dates and setting typical uncertainties for predicted AHe dates
(all information given in ascending retentivity/kinetic population order). The AFT data were recast from QTQt
individual synthetic output files using random spontaneous/induced track (Ns/Ni) ratios that produced central ages
for each kinetic group that were in agreement with forward model predictions using identical EDM parameters with
a ζ-calibration value = 350 yr cm$^{-2}$, induced track density ($\rho_{Di}$) = 2.5 x 10$^6$ cm$^{-2}$, and dosimeter tracks (Nd) = 10000.
These common values made it so each population was simulated as being from the same grain mount for the
purposes of easy comparison and t–T inversion. Population one central age was calculated as: 670 ± 26 Ma,
population two was calculated as: 843 ± 29 Ma, and population three was calculated as: 1602 ± 79 Ma. The
synthetic AFT sample has an overall central age of 934 ± 64 Ma (1σ, $X^2$ = 0.0, MSWD = 9, 34% dispersion, n = 30)
when all age grains are combined. Three mixture model age peaks of 687 ± 34 Ma, 828 ± 34 Ma, and 1602 ± 78 Ma
(1σ) were selected in IsoplotR (Vermeesch, 2018) for the combined AFT data, which are in agreement with the
individual kinetic population central ages. The uncorrected AHe dates used all default RDAAM settings with the
exception of $r_{mr0}$ and the dates were input as: 585 ± 17 Ma, 610 ± 18 Ma, and 819 ± 25 Ma (all 3% errors, 1σ).

We ran QTQt in multiple stages to tune Bayesian sampling and to ensure the acceptance rates for time and
temperature were between ~0.1–0.7, within the acceptable limits discussed in Gallagher (2012). Inversions were run
for >500,000 to >1,000,000 total iterations (burn-in and post-burn-in) and were considered complete when the
likelihood distribution was stationary (i.e., there was no trend in the likelihood values with a stable or "flat" mean;
Gallagher, 2012). The modelling t–T space (prior) was designated as 1000 ± 1000 Ma and 150 ± 150°C with a
maximum allowed heating/cooling rate of 5°C/Myr. Sampling proposed outside of the prior was prevented and more
complex models were rejected. Therefore, t–T points were only added if they provided a better fit to the input data.
The long time interval for these model inversions are styled after a typical cratonic history and the only constraint
that was consistently enforced was starting the model at 300 ± 1°C at 2000 ± 1 Ma. For our purposes, this scenario is
considered a "no constraint" model, since we apply this as a starting condition for all inverse models well above the
sensitivity of our thermochronology data. We also ran models that enforced constraint boxes (i.e., with either one or
two boxes) at 20 ± 10°C at 1650 ± 100 Ma and 20 ± 10°C at 500 ± 50 Ma, requiring t–T paths to pass through them.
These t–T boxes were treated as "known" geologic information for the inversions. For all models presented
hereafter, we show the QTQt Maximum Likelihood (ML; i.e., more complex, best fit t–T path to the observed data,
coloured line) and Expected models (EX; i.e., ~weighted mean ± 95% credible interval; long dashed line and gray
envelope) with respect to the true thermal history used to predict the synthetic data (fig. 1). In Bayesian inference,
the posterior probability is proportional to the likelihood multiplied by the prior, and in QTQt the prior acts as a
penalty against making the model too complex and thus the Maximum Posterior (MP) model will be the simpler t–T
path when compared to the ML path (i.e., typically fewer t–T points; Gallagher, 2012). We have excluded the MP
model for plot clarity for most output because the ML and MP paths are identical or nearly so for most scenarios,
which implies a well sampled and constrained ensemble of solutions (Gallagher and Ketcham, 2020).





**4. Model inversion results**
QTQt inversion results are shown in figure 3 and examine the implications of multikinetic AFT, joint models with
multikinetic AFT and AHe grains using the correct kinetics (i.e., the kinetics implemented during forward modelling
to predict AHe dates), and different combinations of incorrect monokinetic AFT models where the three multikinetic
populations were combined and treated as a single AFT sample and/or AHe dates were assumed to be the
endmember fluorapatite $r_{mr0}$ value. Figure 4 depicts the results comparing observed synthetic data and model
predictions for the inversions in figure 3. The first three models are "multikinetic AFT only" models (fig. 3A–C),
whereas the second row of models depicts results for three multikinetic AFT populations and three AHe grains (Fig.
3D–F). The last three panels are the single population AFT models (fig. 3G–I). We prevented t–T points from being
added during QTQt inversions unless the addition of points provided better agreement between observed and
predicted data. Therefore, all of our preferred results and discussion focus on the Maximum Likelihood model t–T
path, yet we show the Expected model and 95% credible interval for comparison and to provide a general picture of
the overall model ensemble. It should be noted that because the EX model undergoes a simple temperature
weighting in QTQt, the upper 95% credible interval will almost always be biased to slightly cooler temperatures
than if an exponential temperature weighting were to be applied that preferentially weights higher temperatures.
**4.1 AFT-only models – identified multikinetic age populations and correct kinetics**
The first model was setup to simultaneously invert each AFT kinetic population without AHe data for scenarios with
a "no constraint" model, a "single t–T constraint" model, and "two t–T constraints" model (fig. 3A–C). These
simulations were meant to be the ideal case using a lone AFT chronometer with extended thermal sensitivity due to
the presence of multikinetic apatite populations. We investigated the ability of QTQt to recover the true thermal
history using properly identified kinetic age populations while utilizing the true $r_{mr0}$ value from forward modelling
for each population under varying degrees of geologic assumptions or constraints. The general shape, timing, and
magnitude of the true history form and peak temperatures are recovered for the multikinetic AFT models regardless
of whether or not constraint boxes were used. This suggests to us that the combination of high-quality, distinct age
and length populations enhance t–T history resolving power, which becomes progressively improved if kinetic
populations sample a broad range of kinetic space (predicted AFT parameters closely agree with the synthetic data;
fig. 4A-C).
**4.2 AFT + AHe models – consequences of the $r_{mr0}$ parameter**
The addition of the three AHe dates using their *correct* kinetics (i.e., $r_{mr0}$ values) along with the three multikinetic
AFT populations (fig. 3D) improved thermal history recovery with respect to the AFT-only models (fig. 3A–C),
while the addition of two constraint boxes produced a ML model t–T path that reproduced nearly all features of the
true thermal history (fig. 3E). Figure 3E is the best thermal history model that utilized all assumptions and
information used during forward model generation of the synthetic dataset and provides the closest fit to the
synthetic data (fig. 4E). Setting all three AHe grains to 0.83 $r_{mr0}$ produces distortion of the model history with respect
to the true history (fig. 3F). The model predicts three AHe dates that are virtually identical but provide a poor fit to





the input synthetic AHe ages (fig. 4F). The 610 Ma AHe grain (true kinetic $r_{mr0}$ value = 0.83) was on the margin of
acceptability.

### 4.3 Monokinetic AFT models – incorrectly combined kinetic populations

In our experience, multikinetic behaviour is not uncommon for basement samples characterized by complicated
burial histories and nearly always present for detrital apatite samples derived from complex source areas that
experience multiple heating events. In our "monokinetic" scenario, the multikinetic AFT data were incorrectly
treated as a single population and modelled using the central age, MTL, and average eCl or $r_{mr0} \pm 1\sigma$ of the *entire*
*pool* of synthetic single-grain ages. As previously mentioned, combining the three populations caused the sample to
fail the chi-square test ($X^2 = 0.0$) and the calculated AFT central age was 934 ± 64 Ma, the overall MTL was 14.07 ±
1.40 µm (n = 225), and the average eCl is 0.213 ± 0.373 apfu (equivalent $r_{mr0} \approx 0.75$) for all grains. AFT data are
usually treated as such in the published literature and overdispersed data are often modelled regardless of $\chi^2$
statistics. This situation could conceivably occur when the three kinetic populations were either ignored or there was
insufficient kinetic parameter resolution to identify discrete kinetic groups. A sample could also simply not be
multikinetic — but the models here are meant to illustrate the hazards of monokinetic misinterpretation for thermal
history analysis. In the monokinetic simulation without constraints, both the ML and EX t–T paths do not accurately
reproduce the true thermal history (fig. 3G). In this instance the ML path passes directly through both true
Phanerozoic thermal maxima and yields excellent fits to the observed synthetic data (fig. 4G). The addition of two
constraint boxes produced even more complex and highly inaccurate t–T solutions (fig. 3H), yet well reproduce the
observed AFT data (fig. 4H). The AFT sample was modelled as monokinetic again (fig. 3I), but also included the
three AHe dates using uniformly applied default RDAAM $r_{mr0}$ value of 0.83 for each apatite grain to provide further
insight into whether this combination could yield a better outcome just from the addition of more data for the
inversion. The EX model is still inaccurate but the addition of AHe grains made the ML path simpler, nevertheless it
is still distorted and poorly reproduces the true thermal history. QTQt also failed to accurately reproduce the true
AHe dates and predicted the same date for all three grains (fig. 4I). This may be because the second 610 Ma AHe
grain utilized the true $r_{mr0}$ value of 0.83 from the forward modelling and was the best-predicted date of the three
(close to the observed date upper uncertainty limit) and dominated the iterative sampling during the inversion. The
AHe kinetics produced forward model dates that were distinctly older (819 Ma) and younger (585 Ma) than the
(middle) 610 Ma grain but these were unable to be reproduced by the inverse model assuming incorrect $r_{mr0}$ kinetics.



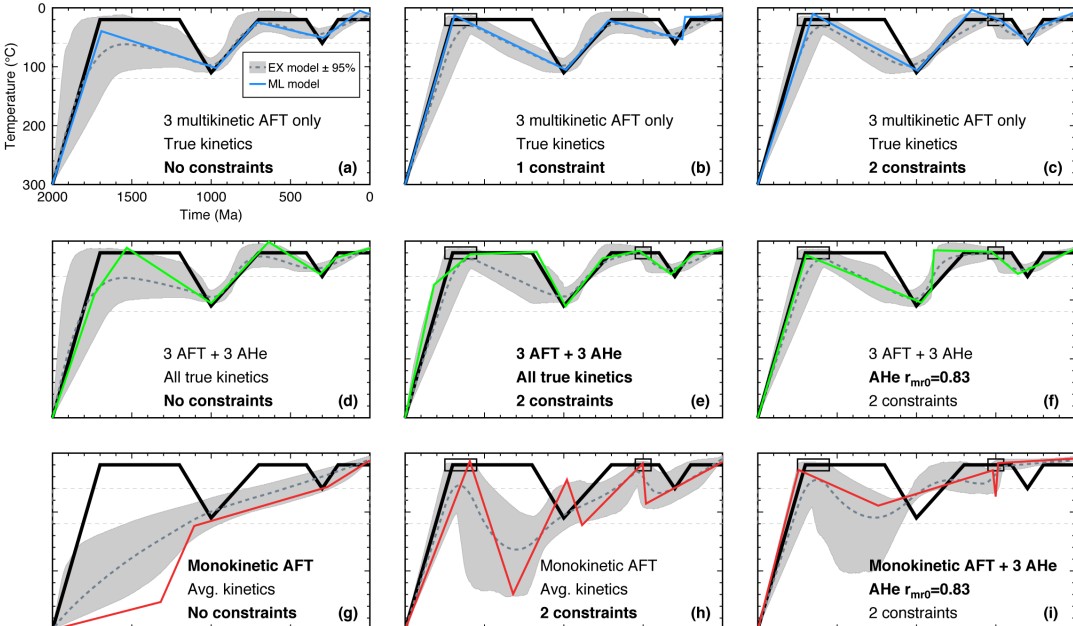

**Figure 3:** Thermal history inversion results from QTQt under different imposed kinetic and t–T assumptions. (A–C) show the "AFT only" models that utilized three multikinetic AFT populations (discussed in the text) as the only input data. The true $r_{mr0}$ kinetics applied during forward modelling were entered in the input files and held fixed for each kinetic population during the inversion. (D–E) show the results of models that correctly utilized three multikinetic AFT kinetic populations and three AHe dates all with the true kinetics held fixed. Panel E is the best model inversion incorporating all correct thermochronometer information used during forward modelling of the synthetic data set. The panel (F) model was completed under the same conditions as panels (D–E) except that the three AHe grains all employ the incorrect (in the oldest and youngest cases) RDAAM default fluorapatite $r_{mr0}$ value of 0.83 as the kinetic parameter. Panels (G–I) were modelled assuming a "monokinetic" or traditional single population AFT sample that combines all three multikinetic populations into one. For all panels: Thick black line is the "true" thermal history from figure 1; coloured, solid lines are the Maximum Likelihood model (best fit) t–T path from QTQt; dashed gray lines are the Expected model t–T path with light gray 95% credible interval envelope. Assumed t–T constraints are black boxes that require thermal histories to pass through them during the inversion.

# 5. Discussion

## 5.1 Apatite composition and multikinetic interpretation

The AFT and AHe modelling results presented here may seem intuitive based on the implemented kinetics and modelling exercises using synthetic data, but are worth discussing nonetheless, since situations where highly variable apatite compositions could influence thermochronometric dates are likely to be encountered in natural samples. The results shown here indicate the benefits offered by interpreting intrasample AFT kinetic populations for inverse modelling and also show how inappropriate assumptions regarding kinetic parameters can greatly influence model outcome. Our examples were determined for a single, distinct thermal history, and yet they establish that apatite composition and multikinetic interpretation (when appropriate) provide valuable information for thermal history modelling — and are mostly unexplored, or at least underutilized by routine AFT studies. Gallagher and Ketcham (2020) also touch on these points in response to the lengthy modelling discussion sparked by Vermeesch and Tian (2014) and are primary themes of this work.





Collection of elemental data and interpretation of multikinetic samples is particularly important for providing greater
t–T resolution (fig. 3A–F), whereas combining or overlooking kinetic populations effectively smears the t–T signal
contained in the individual kinetic groups and produces a meaningless hybrid thermal history model (fig. 3G–I). We
could disregard these incorrect model simulations as self-fulfilling due to forward modelling a synthetic dataset and
assuming "perfect" kinetic models, however for real scenarios we would not know the true thermal history and
without other information, this class of results could be interpreted as geologically meaningful. Perhaps more
important are the broader implications for thermal history modelling if there are inappropriate assumptions
regarding data interpretation and certain steps are not taken to fully evaluate multikinetic AFT samples (fig. 3G–I),
especially at longer timescales where there is greater uncertainty and less geologic control. An important point is
that if multikinetic populations exist and are properly interpreted, they have the potential to constrain a much
broader range of t–T space than an incorrect monokinetic (single population) interpretation for an overdispersed
AFT sample.

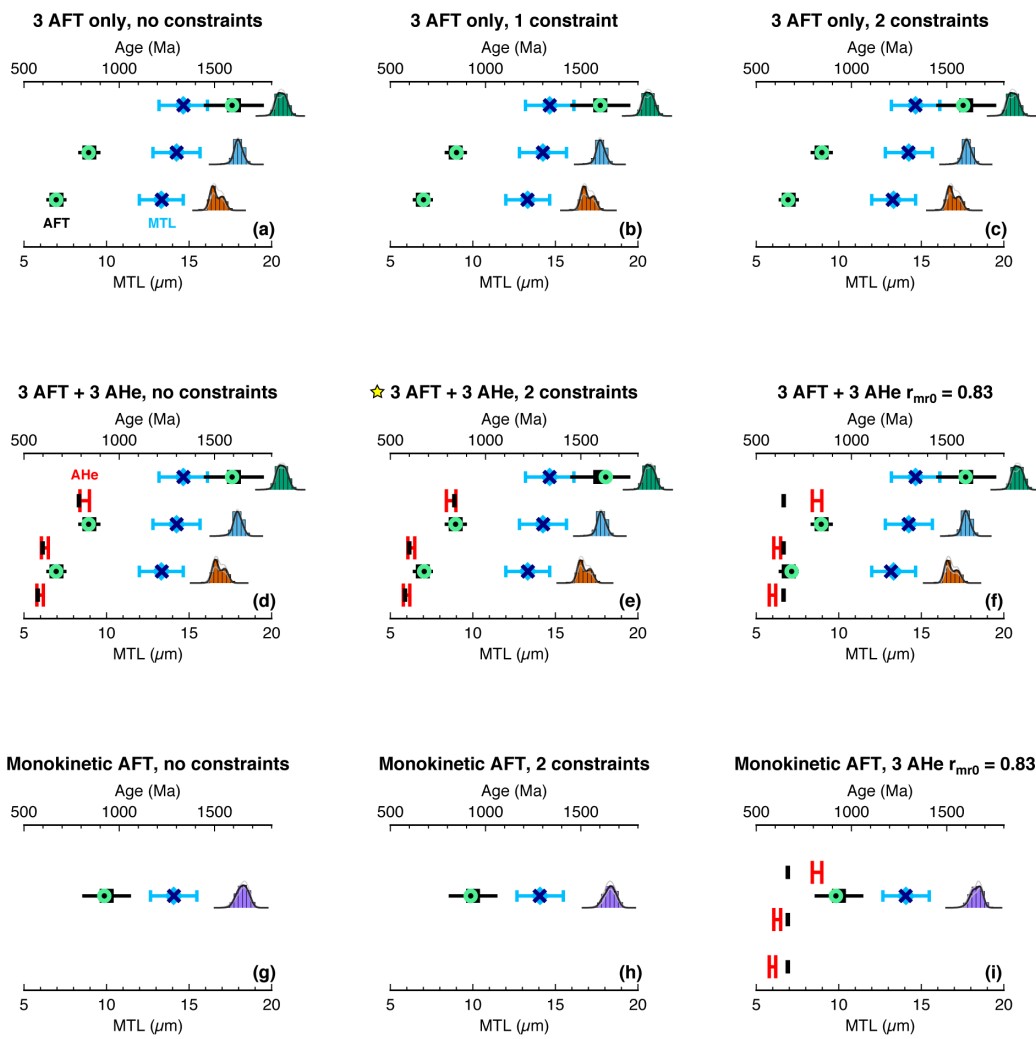

**Figure 4:** QTQt inversion predictions compared to "observed" synthetic thermochronology data generated during forward modelling. Panel letters correspond to counterpart t–T model panels in figure 3. All predictions are for the Maximum Likelihood models. Squares are observed AFT central age ± 2σ, circles are predicted AFT age, diamonds are observed MTL ± 1σ, and X-symbols are the predicted MTL. Individual model fits to each track length distribution for the AFT kinetic populations are also shown and color-coded the same as figure 2. Observed apatite He dates shown by red H-symbol (spans the 1σ error range quoted in the text) and predicted AHe dates are black bars. Panel E with star is our best model that accounts for all multikinetic AFT populations and utilizes the true AHe kinetics and two geologic constraints, all combined for the highest thermal history resolution. Note: track length distributions are arbitrarily placed next to their respective age population and were not plotted with respect to the MTL plot axis.

## 5.2 Data quality and kinetic parameter influence on t–T resolution

The overall temporal and thermal resolution contained in multikinetic AFT data is influenced by multiple factors such as, the amount and distribution of the data (i.e., if the majority of the data are contained in one population versus distributed more equally), thermal history (i.e., the magnitude and sequence of heating-cooling events), and kinetics (i.e., the range of temperature sensitivity). A greater number of different kinetic groups are sensitive to more



extensive parts of the thermal history than a single population. However, the ability to recover thermal history
information depends on the details of the thermal history; if maximum temperatures occur late in the history then
previous events are thermally overprinted and the early history is obscured or erased entirely. We intentionally use
an ideal synthetic dataset with well-defined kinetic populations that have an equal distribution of data across all
populations. Natural populations may have an uneven distribution of grains and therefore populations that contain
the most data will best resolve distinct parts of the thermal history. Our QTQt inversions demonstrate the ability of
these data to inform t–T modelling in the context of variable kinetics and different modeller assumptions. The
similarity between Expected models that do and do not require paths to pass through explicit t–T boxes (e.g., fig.
3A–C) is informative for general modelling practices using Bayesian methods. This tells us that the multikinetic data
being inverted have enough sensitivity to resolve the general t–T history without requiring explicit conditions
imposed on the t–T search. This is perhaps unexpected, as the Bayesian sampling implemented by QTQt generally
favours simpler models over complex ones, which is a possible deterrent for users investigating deep-time thermal
histories (McDannell and Flowers, 2020). However, this should not preclude the use of QTQt for deep-time
problems, as the addition of thermochronological data augments inferences regarding thermal-history complexity in
QTQt.
On the other hand, enforcing constraints while utilizing fewer chronometers and ignoring data complexity or kinetic
trends works against us. The main region of t–T space that proved difficult to resolve in all models was the
prolonged periods at low temperature. This was anticipated since the kinetic models and chronometers themselves
are rather insensitive to temperatures < 50°C. An interesting outcome is that for the "AFT only" models (fig. 3A–C)
the 95% credible interval around the EX t–T path gained precision at the expense of accuracy when constraint boxes
were added (fig. 3B–C), whereas the model without geologic constraints was less precise but more accurate at the
95% level (fig. 3A). It should be noted that since we penalized unnecessary complexity as an explicit condition on
the model prior, there were individual paths (not shown) that were more similar to the true history for these three
simulations, yet QTQt considered these solutions lower relative likelihood. We may expect this compromise
between accuracy (i.e., closer to the true solution) and precision (i.e., greater uncertainty) because subsequent
heating event(s) erase t–T information and the earlier or older, low-temperature parts of the history will be less and
less resolvable with additional reheating and thus may require constraint boxes to assist in the t–T search. However,
imposing constraints where the model is less sensitive leads to exclusion of (potentially viable) solutions and
therefore tightens the envelope of accepted t–T paths. These results suggest that data quantity, quality, the use of t–T
constraint boxes, and proper data interpretation variably trade-off with one another. Figure 3E shows the ideal case
with the most accurate thermal history recovery (nearly identical to the true history) when two constraint boxes are
implemented with three interpreted AFT kinetic populations and three AHe grains modelled using the proper
kinetics. Importantly, this applies in the case of integrating multiple low-temperature thermochronometers and/or
multikinetic AFT data, especially multikinetic populations that progressively diverge in kinetics, therefore
increasing thermal resolution. However, constraint boxes provide no obvious advantage when the three multikinetic
populations are ignored and only the overall central AFT age is modelled (fig. 3H).






We show additional QTQt models in figure 5 to further demonstrate that multikinetic AFT data alone resolve the
true history in the absence of explicit constraint boxes, due to enhanced temperature sensitivity from grain
populations with distinct AFT kinetics. These simulations were carried out to test how robust the inference was for
the "AFT only" models in figure 3 and we show the MP model here for comparison in this case when we are further
exploring resolving power. We ran inversions where more complex models were allowed (fig. 5A; i.e., t–T points
were added even if they did not provide better fits to the observed data), where additional noise was added in the
form of uncertainty on the kinetic parameter value (fig. 5B), and lastly, we ignored the second AFT kinetic
population during inversion (fig. 5C). All other conditions were the same as in previous model runs. At face value, it
seems counter intuitive that we can resolve pre-thermal-maximum temperatures during reheating events because of
how fission-track lengths respond to heating, therefore our resolution should disappear if more complex models are
allowed.

Figure 5A illustrates that in this case there is some resolution lost for the EX model envelope when more complex
models are allowed, in comparison to the EX path envelope in figure 3A. The figure 5A ML path is very similar to
that in figure 3A and the MP path is identical to the ML model. Adding noise to the kinetic data (± 0.05 apfu)
creates little difference between the EX envelopes for the two models for the late history (fig. 5B), signifying a well-
resolved solution ensemble, yet decreased ability to determine the timing and maximum temperature of the second
reheating event and some loss of resolution in the pre-maximum-heating portion of the history. Yet overall there is
not much difference between the figure 3A and figure 5B models, implying that some kinetic uncertainty is not
critically detrimental (under the assumption that our kinetic models are completely accurate, which we know they
are not). The most impactful choice affecting the multikinetic "AFT only" simulations was ignoring kinetic
population two (fig. 5C). The loss of the ~840 Ma fluorapatite AFT population degraded our overall EX envelope t–
T resolution near the peak temperature and affected the ability to recover the timing of the peak temperature. This is
a reasonable consequence because the ~840 Ma AFT age is set upon cooling after the thermal maximum, pinning
the timing and magnitude of heating. The ML and MP models are considerably different and reflect the loss of
independent constraining information within the overall solution pool and demonstrate that without this kinetic
group the "simple" model path is nearly continuous cooling and still adequately explains the observed data.
Accepting more complex models and adding noise led to the loss of t–T resolution, suggesting that our conclusions
are somewhat conditional on the assumptions of preferring model simplicity if there is low signal/noise or fewer
constraining input data. The figure 5C simulation supports the argument that the addition of independent t–T
information (i.e., multiple thermochronometers or AFT kinetic populations) enhances thermal history recovery
without the requirement of constraint boxes.



GEOCHRONOLOGY
Discussions
EGU

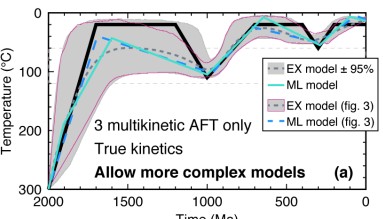 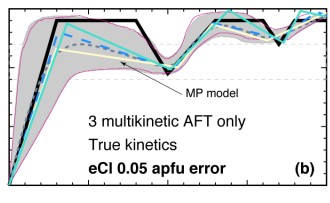 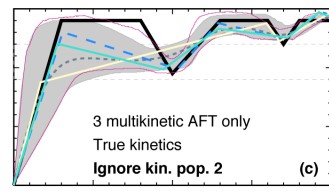

**Figure 5:** Multikinetic AFT-only QTQt models without constraint boxes and same inversion setup as figure 3A–C. (A) QTQt run where more complex models were allowed. (B) Inversion where noise was added in the form of ± 0.05 apfu to the kinetic parameter. (C) Inversion where kinetic population two was ignored. All models: Magenta outline and long dashed blue line are the respective EX model 95% credible interval and ML model path from figure 3A. Thick black line is the "true" thermal history from figure 1; coloured solid cyan and light yellow lines are the respective Maximum Likelihood (best fit) and Maximum Posterior model t–T paths from QTQt for these inversions, short dashed gray lines are the Expected model t–T path with light gray 95% credible interval envelope.

**5.3 The use of constraint boxes in t–T modelling**

The addition of constraint boxes for models with low t–T resolution may yield a false sense of precision in some cases and suggests to us that boxes in QTQt should be used with caution when: (1) thermochronometer sensitivity is marginal or only one chronometer is used, (2) history complexity is presumed to be high, and (3) when histories approach $10^8$–$10^9$ timescales. The use of excessively tiny t–T boxes in QTQt may cause an unintentional linearising bias or artifact to occur because of the Bayesian treatment of user constraints for the prior probability during modelling. Essentially, this means t–T paths may be extremely linear between boxes if more complex models are prohibited. There is also the fact that paths are *required* to pass through a given t–T box. This is especially problematic for geologic histories involving unconformities where, for example, we know basement rocks were at surface by 450 Ma because there are preserved sediments of that age nearby. However, this information does not preclude samples actually being exhumed close to the surface at 650 Ma and sitting at or near the surface for 200 million years before the deposition of Ordovician sediments. During Bayesian modelling, undue influence on the t–T search may occur if a constraint box were implemented at 450 ± 10 Ma (depositional age) and 10 ± 10 °C (surface temperature). We may suspect an issue if the majority of thermal histories showed a very linear, preferred t–T segment through our constraint box, yet some more complicated histories with a greater number of t–T points (i.e., penalized more complex paths) were also visible yet exhibited cooling prior to our box constraint, albeit with less frequency. Unfortunately, under random Monte Carlo modelling assumptions, this "box biasing" would never be recognized as a problem due to the reliance on boxes for informing and expediting the t–T search. It is nonetheless difficult to generalize the use of constraint boxes for inverse modelling and outcomes ultimately depend on how constraints are implemented. Consequently, it is important to simulate and report several scenarios using different explicit conditions for the t–T search for complex histories (e.g., Gallagher and Ketcham, 2018).

When confronted with using one or two low-temperature thermochronometers over longer timescales, the choice that is often made is to add more t–T boxes to better delineate the model space. However, this opens the door for assumptions to be heralded as geologic evidence, and as we see in these examples, this can still yield inappropriate thermal histories when t–T resolution is low. It is important to differentiate between geologic constraints (e.g.,



stratigraphic relationship or basement nonconformity) and assumptions (e.g., regionally rocks cooled below
$^{40}Ar/^{39}Ar$ biotite closure temperature of ~300 °C at ~2000 Ma) and to test different scenarios during modelling.
There are obviously exceptions to these points but this simply means that a model outcome is only as good as the
input data, and that tackling a complex problem with high expectations and few data should not — and cannot —
result in exceptional model results without numerous assumptions and choices made by the modeller. That being
said, we disagree with the recent assertion by Green and Duddy (2020) that *"thermochronology data in isolation*
*cannot define periods when samples were cooler and subsequently reheated. This can only be defined with the aid of*
*constraints from geological evidence."* This statement alludes to the non-uniqueness of t–T models and applies in
situations where a single AFT age population is modelled, or more generally when only one thermochronometer is
used to elucidate complicated t–T histories. However, we propose that multikinetic AFT interpretations (or more
generally, integration of independent information from multiple chronometers) demonstrate that their view does not
always apply, as we can see illustrated in figure 3A. Green and Duddy (2020) also go on to state that slow,
continuous cooling is often assumed in published thermal history models and that this is inappropriate. Of course,
ignoring geologic information and blindly inputting thermochronology data into modelling software will always
yield inappropriate thermal histories — and there is nothing preventing the user from doing this. However, model
simulations such as the one that we show in figure 3G tell us that the wrong model may *imply* slow monotonic
cooling, although it is not outright assumed, whereas our examples that utilize high-quality data (fig. 3A–E)
demonstrate that universal slow cooling suppositions are invalid.

**5.4 Thermochronometer kinetics and future considerations**

Our modelling results reveal that attempts to understand or interpret latent multikinetic age populations within AFT
data provide meaningful information that makes sense within the appropriate kinetic context of both the AFT and
AHe thermochronometers. This demonstrates that excess age scatter for AHe dates may be governed solely by
composition (i.e., $r_{mr0}$) for grains of the same morphology and U content — and in this case, individual AHe dates
may be older than an AFT central age for valid reasons. These t–T models based on synthetic data reinforce the
conclusions of Gautheron et al. (2013) that specifically examined the track annealing law with respect to apatite
grain chemistry ($r_{mr0}$) and the effects on AHe dates. Our results are even more telling when one considers that the
elemental data required to calculate $r_{mr0}$ are often not collected and the typical AHe fluorapatite ($r_{mr0} = 0.83$)
assumption can be misrepresentative for t–T modelling and produce inaccurate thermal history results, which was
recently demonstrated with natural samples by Recanati et al. (2017) and Powell et al. (2020). This implies that
radiation damage effects on He diffusivity are only one piece of the "kinetic puzzle" and that serious t–T
inaccuracies can propagate into thermal history modelling if apatite composition, and therefore kinetics, are
incorrectly determined or simply assigned as some default value for both the AFT and AHe methods. The distorted
thermal history results shown here for the commonly used RDAAM (assuming uniform $r_{mr0}$) support the adoption
and routine use of a He kinetic model independent of fission track annealing kinetics (Gerin et al., 2017; Willett et
al., 2017). This is especially applicable to deep-time histories where rocks have greater potential to undergo cyclical
burial and exhumation because there is a greater age discrepancy introduced by the RDAAM when rocks spend



extended time at temperatures < 80 °C (Willett et al., 2017). Another conclusion is that, in the case of examples such
as figure 4F (or fig. 4I), the poorly predicted AHe dates are a result of an incorrect kinetic assumption and have
nothing to do with the quality of the dates. In many cases, in light of the known age scatter issue surrounding the
AHe method, workers may suspect these dates are erroneous because t–T modelling is not exhibiting agreement
between observed and model dates. We stress that this disagreement could also be a result of incorrect model or
geologic assumptions and have less to do with the observed age data, which further emphasizes an incomplete
understanding of kinetics in our models. The implications of our modelling exercises would also advise against
attempts to arbitrarily cull AHe datasets of "outliers" and recommend this should be carried out only in extreme,
obvious cases of internal date disagreement or where other evidence is brought to bear on the source of excess age
scatter, such as analytical screening of apatite degassing behaviour (Idleman et al., 2018; McDannell et al., 2018).
Some of the examples discussed in McDannell et al. (2018), such as the Sierra Nevada apatite suite, exhibit similar
grain sizes, U content, and CRH degassing behaviour nearly identical to Durango apatite (the common laboratory
age standard), yet intrasample age scatter persists. Those experiments established that there are still variables that
affect He diffusion that are difficult to characterize, such as the siting of He in the crystal lattice, how radiogenic He
diffuses or is trapped during the transition from an open-to-closed system, and the specifics of how He is liberated
during laboratory heating. This persistent age scatter may be partly explained by differences in He retentivities (i.e.,
grain chemistry) that were unable to be documented directly by degassing patterns alone in those experiments.

We advise against the growing practice of trying to smooth out He date dispersion by arbitrarily binning grains into
groups by effective uranium (eU, parent U, Th, and Sm content weighted for α productivity) that can be represented
by an average date (cf., Anderson et al., 2017; Anderson et al., 2018; Weisberg et al., 2018). This presupposes that
the age dispersion is caused by analytical uncertainty rather than geological factors such as variable composition and
radiation damage, even when we know the latter to be the prevailing sources of scatter. Although binning may make
it easier to fit the data using simple model assumptions, it can potentially lead to distorted, close-fitting thermal
histories that are interpreted as good solutions. Such an approach can act as a disincentive to acquire the data needed
to better understand the causes for the age dispersion. Ultimately more research is necessary to address this issue
but, in the absence of definitive data, it is worth trying to determine the model parameters that are needed to fit the
observations. The advantage of Bayesian models like QTQt is that kinetic parameters are acknowledged to be
uncertain and attempts are made to accommodate complicated thermochronologic datasets by adjusting kinetic
parameters within specified uncertainty ranges. This is preferable to rejecting datasets that are deemed incompatible
because they cannot be reconciled using simple modelling approaches with invariant parameters.

Currently there are limited options for directly quantifying kinetics for AHe grains outside of traditional (U–Th)/He
step-heating or CRH diffusion experiments. However, the maturation of in situ laser ablation (U–Th)/He (Boyce et
al., 2006; Pickering et al., 2020) provides a non-destructive micro-analytical method (at the expense of diffusion
information) that yields age information and may improve sample characterization by allowing auxiliary elemental
analysis. A way to bridge the gap and provide insight into possible chemical heterogeneity of unknowns is to carry



out AFT analyses and have grain mounts characterized by electron microprobe to obtain apatite elemental data while these methods are further developed and adopted. This approach will give a first-order approximation of the spread in apatite chemistry for aliquots analyzed for (U–Th)/He (e.g., Powell et al., 2020). We envision coupling laser ablation inductively coupled plasma mass spectrometry (LA–ICP–MS) AFT and AHe double-dating as a methodology that will prove valuable for better characterization of apatite chemistry in relation to derived dates, which will inform and require future laboratory experiments to be conducted on a diverse suite of apatites to better constrain annealing behaviour and extrapolate it to geologic timescales.

We recommend the routine collection of elemental data for apatite dated using the fission track method as a means to better quantify sample chemical variation and relate this to kinetic behaviour for thermal history analysis. The use of $r_{mr0}$ (or eCl), while imperfect, still provides the best resolution for kinetic interpretation. The use of other kinetic parameters such as $D_{par}$ are generally inadequate for more high-level kinetic interpretation due to low precision (Issler et al., 2018; McDannell et al., 2019b; Schneider and Issler, 2019), whereas the sole use of measured Cl is only applicable for a limited suite of apatites (i.e., chlorapatite) and neglects the influence of other elemental substitutions on annealing (i.e., Carlson et al., 1999; Barbarand et al., 2003). These topics are discussed more fully in a future companion paper that examines detrital AFT samples from Yukon, Canada to illustrate multikinetic AFT interpretation and modelling methods.

## 6. Conclusions

Using synthetic data derived from forward modelling, we show that, under ideal conditions, it is possible to extract multi-cyclic heating and cooling history information from multikinetic AFT and AHe data using inverse modelling methods when kinetic parameters for AFT annealing and AHe diffusion are correctly specified. Essential details of a two-phase heating and cooling history are reproduced using AFT multikinetic data alone without imposing constraint boxes but the closest fit to the true solution is achieved using all the synthetic data with constraint boxes. Alternative monokinetic interpretations that ignore multikinetic behaviour generate solutions that significantly depart from the true solution while providing close fits to the reinterpreted AFT data; under these conditions, imposing constraint boxes can make the t–T solutions worse. Recent publications suggest that composition can influence He diffusion in apatite; in the context of our simulations, ignoring composition causes misfits to AHe dates and degrades model thermal histories. These results suggest that apatite elemental data should be acquired for interpreting and modelling overdispersed thermochronological datasets that result from multikinetic AFT annealing and AHe diffusion behaviour. The ability to recover high-resolution thermal histories from natural multikinetic AFT samples depends on the details of the thermal history and characteristics of the data and this is the subject of a future paper.

## 7. Appendix

The appendix contains the true thermal history and the synthetic AFT data set. See the main text for further details.





**8. Author Contributions**
KTM designed research and performed modelling. DRI was involved in conceptual discussions, model evaluation,
and editing and drafting the manuscript. KTM wrote the paper.
**9. Competing interests**
The authors declare that they have no conflict of interest.
**10. Acknowledgements**
The authors graciously thank Kerry Gallagher for making changes to QTQt specifically for these modelling
exercises. KTM also thanks his partner Jennifer for all of her help during the writing of this manuscript.

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

**Figure Captions**
**Figure 1:** Thermal history used to predict synthetic AFT and AHe data. This t–T path is referred to as the "true" thermal history
throughout this paper. The predicted synthetic data were then used as input for QTQt to recover the thermal history through
inverse modelling. PAZ = partial annealing zone for fission tracks.
**Figure 2:** Predicted synthetic AFT data from the thermal history in figure 1. Multikinetic age populations were individually
predicted using distinct $r_{mr0}$ kinetics shown in (B) panels (discussed in the text). These data were then input in QTQt and inverted
in an attempt to recover the true thermal history in figure 1 (see fig. 3). **(A)** Central age and 1σ errors are indicated for each
kinetic population. Kinetic populations one, two, and three are displayed as arms on their respective radial plots, with individual
AFT ages closer to the origin being less precise. The last radial plot shows all thirty individual grains and demonstrates that when
taken together, the combined sample fails the $\chi^2$ test ($p < 0.05$) for homogeneity (i.e., that all grains belong to a single underlying
age population) suggesting multiple age populations. This is the scenario most researchers would start with before evaluating the
sample for potential multikinetic behaviour. Mixture modelling was subsequently performed on the combined sample and the
model age peaks that were picked seamlessly align with the individual kinetic population central ages. This aligns with how
populations would be defined and compared with the elemental chemistry for individual age grains during multikinetic
interpretation. **(B)** The predicted track length distributions for each kinetic population from the thermal history in Figure 1 using
the specified kinetic parameter value. The last panel on the right combines all tracks from each kinetic population. Numbers on
the histogram are the number of tracks in each μm bin. Abbreviations: eCl = effective Cl; MTL = mean track length.
**Figure 3:** Thermal history inversion results from QTQt under different imposed kinetic and t–T assumptions. (A–C) show the
"AFT only" models that utilized three multikinetic AFT populations (discussed in the text) as the only input data. The true $r_{mr0}$
kinetics applied during forward modelling were entered in the input files and held fixed for each kinetic population during the
inversion. (D–E) show the results of models that correctly utilized three multikinetic AFT kinetic populations and three AHe
dates all with the true kinetics held fixed. Panel E is the best model inversion incorporating all correct thermochronometer
information used during forward modelling of the synthetic data set. The panel (F) model was completed under the same
conditions as panels (D–E) except that the three AHe grains all employ the incorrect (in the oldest and youngest cases) RDAAM
default fluorapatite $r_{mr0}$ value of 0.83 as the kinetic parameter. Panels (G–I) were modelled assuming a "monokinetic" or
traditional single population AFT sample that combines all three multikinetic populations into one. For all panels: Thick black
line is the "true" thermal history from figure 1; coloured, solid lines are the Maximum Likelihood model (best fit) t–T path from
QTQt; dashed gray lines are the Expected model t–T path with light gray 95% credible interval envelope. Assumed t–T
constraints are black boxes that require thermal histories to pass through them during the inversion.
**Figure 4:** QTQt inversion predictions compared to "observed" synthetic thermochronology data generated during forward
modelling. Panel letters correspond to counterpart t–T model panels in figure 3. All predictions are for the Maximum Likelihood
models. Squares are observed AFT central age ± 2σ, circles are predicted AFT age, diamonds are observed MTL ± 1σ, and X-
symbols are the predicted MTL. Individual model fits to each track length distribution for the AFT kinetic populations are also
shown and color-coded the same as figure 2. Observed apatite He dates shown by red H-symbol (spans the 1σ error range quoted
in the text) and predicted AHe dates are black bars. Panel E with star is our best model that accounts for all multikinetic AFT



populations and utilizes the true AHe kinetics and two geologic constraints, all combined for the highest thermal history
resolution. Note: track length distributions are arbitrarily placed next to their respective age population and were not plotted with
respect to the MTL plot axis.
**Figure 5:** Multikinetic AFT-only QTQt models without constraint boxes and same inversion setup as figure 3A–C. (A) QTQt run
where more complex models were allowed. (B) Inversion where noise was added in the form of ± 0.05 apfu to the kinetic
parameter. (C) Inversion where kinetic population two was ignored. All models: Magenta outline and long dashed blue line are
the respective EX model 95% credible interval and ML model path from figure 3A. Thick black line is the "true" thermal history
from figure 1; coloured solid cyan and light yellow lines are the respective Maximum Likelihood (best fit) and Maximum
Posterior model t–T paths from QTQt for these inversions, short dashed gray lines are the Expected model t–T path with light
gray 95% credible interval envelope