# Peer review of "Simulating sedimentary burial cycles: I. Investigating the role of apatite fission track annealing kinetics using synthetic data"

_Geochronology, 2020_

## Referee Comment (RC1) · Kerry Gallagher (Referee) · 13 Dec 2020

Simulating sedimentary burial cycles...

McDannell and Issler

This manuscript presents a series of modelling results to demonstrate that use of compositionally defined subpopulations of apatite fission track (AFT) data can provide more detailed information on reheating events in protracted (deep time, 1 Byr) thermal histories than would be intuitive based on the general understanding of annealing of fission tracks. I found the paper a bit long and wordy, even flowery at times - a bit stream of consciousness sometimes - and the wood is getting lost in the trees. Having said that, this review may suffer from similar meanderings.

The paper falls in the scope of GChron, presents new ideas and demonstrates the utility of detailed compositional data for modelling AFT data, on the assumption that the calibrated annealing models are OK.

Perhaps the authors could try to reduce and reorganise the text to help the reader. For example, 2.5 pages on rmr0 calibration is a bit of a distraction - much of that could go into appendix/supplementary (perhaps keep something on effective Cl for the main text). There is a lot of detail from the forward model (predicted mean lengths, initial lengths) in the text that is also on the figures, so just keep the latter, or put all the numbers in a table (and the table perhaps in supplementary ?). Also, the results of using the wrong model (e.g. mono-compositional when it should be multi) are probably too long. I think most of us would appreciate that using the wrong model is likely to be a problem. The important point may be that we can still fit the data reasonably well (using a single sample).

They need to state clearly up front the assumptions underlying some of the models - for example the multi-element/compositional models for the calibration to rmr0 or effective Cl (eCl) are not perfect, are likely to contain correlations between the fitted parameters). If they do not have access to the original data or calibrations, then perhaps some kind of resampling could done (e.g. take some elemental composition data, resample those data using typical uncertainties and recalibrate the model). Also, their example (synthetic data) are very clean and distinct in their compositions. Do we see/expect such well separated populations often, and if so how have these been dealt with previously ? When does the ability to resolve the thermal history based on compositional groups start to deteriorate if the compositional groups are less distinct ? Going to the extreme, the conclusion that we might draw from this study is that we should model each grain with its own specific compositionally defined annealing model (and model parameters). I agree with the authors that we often need to consider sub-populations of data from a given sample both for AFT and AHe, and that averaging the data prior to modelling is probably not a good idea (or at least we need to acknowledge that we will obtain some kind of average, perhaps unrepresentative, solution and that we are potentially throwing out information). However, the other side of the argument would propose that the predictive models are not that sophisticated, not free of uncertainty, and not even really based on a well developed understanding of the physical processes and how they operate on geological scales.

Overall I think that main premise could be demonstrated more efficiently. The idea is that chemical composition (and perhaps associated mineral structure changes) has a major effect on annealing and diffusion in apatite, and this effect is multi-element, rather than just Cl/F as sometimes assumed for fission track annealing. It is a good idea to promote an analytical protocol of measuring a wide range of elements, rather than just Cl, or a proxy such as Dpar, as these data may be useful in future for annealing model recalibration and/or provenance (e.g. O'Sullivan et al. Earth Science Reviews 201, 2020). However, the available model calibrations are not based on a lot of data, and as stated in Carlson et al. (1999) "*in the absence of any physical understanding of why compositional variations impede or enhance annealing, we have little confidence that it can be used meaningfully to predict the annealing behavior of apatites not included in the experiments*". The concern is that these preliminary calibrations are assumed to give us the definitive model, free of uncertainty, and the rather strong conclusions about the inference of thermal histories are based on that assumption.

For example, the results presented are based on more or less ideal data with well separated (kinetic/composition and age) populations (as described in 236 to 249). In this case, we can pretty much recover what we started with including heating out needing to specify near surface constraints on the thermal history. Furthermore, the modelling approach implemented in QTQt tends to prefer simple models, conditional on fitting the observed data adequately relative to more complex models. I think this should mean that individual models making up the credible interval range figure 3a will tend to look like the ML model shown in that figure, and we do not fully recover the thermal history of 20°C for about 500 m.y. prior to the first heating event ) - I would guess it is just the timing of reheating that changes, and probably the same for the second heating event.

The authors then demonstrate that combining these sub-populations and assuming an average composition (but generally fixed) leads to lack or resolution and/or spurious results for the inferred thermal history. The authors often imply that the latter is common practice, but do not really give any concrete examples. I think many, if not most, people working with fission track data are aware of the potential for over dispersed age data and hopefully would deal appropriately with an over dispersed population (using subpopulations based on composition or age, or perhaps remove egregious outliers - also this issue seems to have different significance depending on whether the data are collected with the traditional EDM method or LA-ICPMS, the latter method tends to have greater dispersion but similar central ages.

Some general small points

The idea of AFT ages increasing then decreasing with Cl content was mentioned some time ago, I think by Barry Kohn, who had some data from Canada implying that the age decreases at high Cl (6% ??) which was associated with a change in crystallographic system from hexagonal to monoclinic. Not sure he ever published that though.

Not clear if the sensitivity of the data to composition as proposed is enhanced due to the long timescales, or this is a general feature.

Also you switch between different combinations of AFT and AHe data, but the latter are fairly minor to the main proposition (and their sensitivity is based on the radiation

damage model based on a FT annealing model). You could remove the AHe aspect totally and not change the message...expect perhaps in section 5.4, which comes pretty late and out of the blue.

Specific comments

Lines 197 (and 213)          There seems to be the assumption that AHe ages depend on composition, but is this just because the assumed model for radiation damage is one incorporating a fission track annealing model

Line 214 - QTQt will generate thermal histories regardless of ..data.... This is true of any sampling based modelling approach. The important point is how the generated thermal histories are accepted or not - QTQt effectively uses the ratio of data fit  (likelihood) between a current and proposed model, while HeFTy, another piece of software for modelling thermal histories from thermochronological data, uses an absolute approach (p-value as a measure of data fit) for each thermal history. Perhaps accept is a better word than generate (also on 218), but keep the statement about the user needing to assess the output, particularly how well any particular model predicts the observed data.

Line 249 - 3% seems small for AHe age data ??

Line 256 - what happens if you do not constraint the heating/cooling rates ?

Line 257 - "t-T points were only *accepted* if they provided.."..the points are added, but the important step is whether they were accepted or not.

Line 262 - put the constraint definitions in section 3.1, and perhaps explain what they represent geologically.

Line 264 - the ML model is potentially more complex than the MP model, but not always, and similarly the MP is not always simpler than the ML model...they can be the same.

Line 285 - not sure what you mean by a simple temperature weighting....the expected model is defined as

$$E(x) = \int x \, p(x) \, dx$$

and in this case the p(x) is the posterior. Given that the distribution of accepted models is the posterior then we just take the arithmetic mean of all accepted models to do that integral. The lower temperatures are because the distributions on temperature around the time of maximum temperature are often skewed (to lower temperatures) and so that leads to lower values for the expected (mean) maximum temperature. However, we often see that the duration of time close to the maximum temperature is greater for the expected model than say the ML model. This may tend to compensate a little in terms of fitting the data, but often not enough...you need to look at the predicted values relative to the observations. Note that we do not generally expect sharp V shaped thermal histories anyway (due to diffusion), but that is another issue.

Line 314 - QTQt does not use the central age directly (or even indirectly) as a model constraint. The predicted age for a given kinetic parameter and thermal history is used

to infer the equivalent predicted $\rho_s/\rho_i$ ratio which is then used in the likelihood function with the measured $N_s$ and $N_i$ values for each grain (see Gallagher 1995, EPSL).

Section 4.2 -line 300  perhaps just discuss using the correct kinetics (here you choose rmr0, but this could be another parameter, even if less sensitive ?)

Line 317 - when using the average eCl 0.213±0.373 in QTQt, this implies that you let the kinetic parameter vary as part of the inevrsion...what was the distribution of the accepted values ? If skewed to a higher or lower value, that may be informative concerning the sensitivity of the different subgroups of data to an average/common kinetic parameter.

Line 318 - perhaps some examples of overdispersed data treated as a single population The impact of this is likely to depend on how overdispersed and why...failing the chi-sq pvale = 5% test is not necessarily the definitive indication (e.g. we can pass at a level of 5.0001, but fail at 4.9999). Leaving aside analytical problems, dispersion that is a real if sometimes unwelcome signal could be due to compositional effects and discrete provenance related age populations (for which compositional ranges may be similar).

Line 320 - we do not necessarily have to formally identify discrete groups with mixure modelling, but just divide compositional range into subgroups and use the appropriate values (e.g. as Geotrack seem to do for Cl binned at 0.1wt % intervals)

 Line 330 - it is not that QTQt failed to reproduce the true AHe dates...it is because the wrong choice of model prevented QTQt from doing so....

Line 359  - I think that Geotrack do use compositionally discrete modelling for their routine AFT studies, but we rarely get to see the predictions for their preferred models.

Line 388 - data quality is important too...

Line 391 - what do you mean by more extensive ?

Line 396 - as I said above, perhaps we should model each grain with a specific set of kinetics ? This does no necessarily require running N annealing models for N grains, but perhaps 4-5 and we can interpolate the results (e.g. predicted ages and length distributions) for intermediate compositions.

Line 401 - without necessarily requiring

Line 407 - what was the first hand ? I had forgotten by the time I got here.

Line 413-415...not sure I understand this...explicit condition on the model prior ? Perhaps you mean on the model sampling.

Line 415 - not correct...it is the posterior that will be lower I think, rather than the likelihood. At least the acceptance of models is based on the posterior

Line 418 assist - use focus perhaps.

Line 421 - proper data interpretation is vague...you mean some of sub-population classification based on composition, age dispersion ?

The discussion of precision v accuracy based on the credible intervals does not really take account of the form of the individual thermal histories...although they may define an envelope that seems consistent with the known true answer, the actual thermal histories may not really capture the true thermal history as well.

Line 429 - again, it is difficult to assess if we really resolve the true thermal history from the credible intervals..they do not tell us about the variation of indiviudal thermùal histories...so the earlier reheating event may start at different times, but we do not necessarily resolve the period sitting at low temperatures for a long time.

Line 434 - more complex thermal histories were accepted or reatined rather than added.

The rest of this paragraph to line 439...when you allow more complex models that do not improve the data fit to be accepted, you start to sample the prior more extensively, i.e. you tend to fill up the prior box in those parts of the time-temperature space where the data do not care what happens. For example, prior to a reheating event, the temperature can be pretty much anywhere from the reheating maximum value to the lower limit of the temperature prior. This does not mean we resolve the thermal history there. Almost the opposite..all we resolve is that the temperature has to be lower than the subsequent maximum.

Line 443 - Apfu Cl ?..if you add normally distributed noise, or even uniformly distributed, with the same mean value as the true model, it is not a great surprise that the expected model and credible intervals do not change much as we will tend to fit the average values of the data (which will reflect average values of the kinetics, which will be sampled on average to give the correct value...if that makes sense).

Line 448 - this acknowledgement of imperfect kinetic models should be stated much earlier, and ideally quantified somehow..(i.e. the recalibration exercise I mentioned above).

Line 456-58 - the data quality (and the associated information on the thermal history) is first order in terms of what we can resolve. If the data are poor, scarce, we want more uncertainty.

Section 5.3 - this deals with a separate issue to the main thrust of the paper, and adds to the feeling of stream of consciousness. I think it is not just with QTQt that we need to assess the effect of constraints on the final results and this has been said before in the exchanges in Earth Science Reviews over the last few years, and still ongoing. Any model result is conditional on the assumptions made to obtain the result. Thus a no constraint model can often be relatively boring (i.e. linearish cooling) but if the data can be adequately explained that way, then it is useful to know and can be considered as an end member model. Adding constraints is not a problem, but these need to be justified and this needs to be clearly stated in any study, preferably with some assessment of the confidence in a constraint. Additionally, when a thermal history is composed of linear segments joining up constraint boxes and we fit the data, this does not mean the

constraints are justified...just that the data do not contradict these constraints and they do not require more complexity than imposed by the constraints.

A combination of inverse models and targeted forward models, often based on the inversion results, can be useful to deal with many of the problems discussed in this section and you do say that ..but you could just say that more concisely.

Lines 474-476...not clear what this means...especially linearizing bias and Bayesian treatment of user constraints ..?

Line 494..I agree with the sentiment...heralded perhaps better stated as imposed under the guise of geological evidence. This goes back to what I said above - there is not problem using constraints/forcing a model, but the results are conditional on these and there are generally other models that can fit the data without, or with different constraints. Again, it is how valid the constraints are that is the big question.

Line 504...again Geotrack seem to use 0.1 Cl wt% bins when modelling, but we never see their predictions. You may add here that limitations arise when modelling samples independently, especially in boreholes. It is clear that jointly modelling multiple samples (plus the mutli-kinetic approach for each sample) is better - data noise will tend to cancel out (if it is random) while real signal (the thermal history) should be reinforced. See Gallagher et al EPSL 2005 for an example with synthetic data too...

Liner 509-510...what if the true thermal history is simple cooling ?

Line 513- not sure what you mean by universal slow cooling suppositions...there is nothing forcing cooling in QTQt, apart from perhaps have a sample at surface temperature today and letting it start hotter...if the data are happy with that, then why not ? If the data need more complexity the thermal history should adapt.

Much of section 54. is relatively speculative but more importantly deviates from the main message of the paper concerning FT annealing and left me a little confused. The last paragraph is OK...but I think you could drop much of this and perhaps save it for another paper....but that is up to the authors. While I agree that allow an effective parameter to vary in the modelling, as QTQt allows, can be useful - the paper by Ricanati et al. demonstrated that yes we can improve the fit to the data by having a wiggle factor for each grain diffusivity. As Samuel Karlin said, 'The purpose of models is not to fit the data but to sharpen the question'...so demonstrating we can fit the data with an additional factor is not really the solution, but suggests we should look for a physical control on that factor.  Given the arguments here, I would say go and measure apatite chemistry to demonstrate that there is some correlation of age and chemistry or effective diffusivity then you are on to something  (perhaps this will be possible with the method, Pickering et al. mentioned). For me, as it is, the sceptic will take this section as special pleading for a control on AHe date dispersion that is neither understood nor constrained. Those adopting the averaging strategy for AHe data will just say we do not understand enough to do anything more sophisticated and carry on. I would add a caveat to the Karlin quote too....not fitting the data means we need to ask different questions.

Line 582...Steve Bergmann was often insistent on the importance of OH as a control on fission track data....not sure if he ever published anything though.

Line 594 - what recent publications ?

Line 595 - not misfit, but poor fits.

Line 596 - degrades in what sense ?

It may be the paper may be better concluded by adding a series of recommendations on analytical practice/protocol and then modelling strategies.

Fig 1 - perhaps put the 2 constraint boxes on the forward thermal history. They are a little strange as constraints as we might expect the constraint to be the stratigraphic age of sediment deposited at the time of the start of the heating events, rather than some time prior to the heating event.

Fig 3 - the credible interval ranges are fine as presented in this figure, but it may be useful to put the sampling of individual models and/or the marginal distribution (the coloured plot from QTQt for a given sample thermal history) in the supplementary - then we can see how many models actually start reheating at 1200 Ma for example.

---

## Referee Comment (RC2) · Anonymous Referee #2 · 15 Dec 2020

"Simulating sedimentary burial cycles: Investigating the role of apatite fission track annealing kinetics using synthetic data" by McDannell and Issler explores the potential of multikinetic AFT data to improve the resolution of time-temperature paths. In its simplest form, the paper argues that by increasing the number of thermochronometric systems sensitive to different temperatures, thermal histories are easier to infer. Here, each kinematic population could be viewed as a different system with slightly different sensitivity. This concept is reasonably well understood, however, these sorts of analyses are required given the recent discussions around thermochron reliability. The study is suitable for publication with minor revisions but it could be substantially shorter. The paper basically boils down to resolution tests. It has been previously pointed out that

the multikinetic data improve the quality of thermal histories and this paper builds on that point. However, a resolution test is only really valid for the specific test at hand. It is clear that by using all the data in the correct way, the correct path can be recovered for this specific path, but this does not mean that multikinetic data is important for all time-temperature paths. This is an obvious problem with this type of study and unless the resolution tests are for a specific problem at hand, it is unclear what we really learn from these tests. For example, if the burial conditions were ever so slightly different, the tests may imply that AHe data are more important that AFT data or something like that. This point should be acknowledged and maybe the details of why the forward model path is as it is should be discussed. It is also unclear whether we are learning something about the data or something about the specific algorithm used to interpret the data. One of the attractive things about QTQt is that you do not need to specify the number of nodes in the inversions or limit the rate of the cooling. However, the authors stress the importance of the maximum likelihood models and not the expected or maximum posterior models that benefit from the reversible jump component of the algorithm. No Bayesian statistics are required to find a maximum likelihood model, so it is unclear why QTQt is used. For example, in Figure 3e, the ML model is actually outside of the credible intervals around the EX model. This is probably an advantage of the EX model because often the ML model is wildly complex. If the EX models are taken as a suitable compromise between averaging data and resolution, all of the first 6 models in figure 3 (3a-3f) look very similar. This highlights some of the points made by Vermeesch and Tian (2014) and some discussion about which model to concentrate on might be useful. An additional set of models that use predicted data using one kinetic population and then invert them with the correct single population kinetics might help demonstrate the value added in having samples with multikinetics. I also think that in many cases detrital samples have multikinetic data because the apatites are from different source areas. In turn, there is the basic assumption that it is appropriate to treat all these crystals as having the same thermal history. In fact, the crystals may have distinct thermal histories that may or may not be important in the interpretation.

For example, Carter and Gallagher (Carter, A. and Gallagher, K., 2004. Characterizing the significance of provenance on the inference of thermal history models from apatite fission-track data-a synthetic data study. SPECIAL PAPERS-GEOLOGICAL SOCIETY OF AMERICA, pp.7-24.) describe this issue for the case of AFT data and Fox et al., 2019 (Fox, M., Dai, J.G. and Carter, A., 2019. Badly behaved detrital (U‐Th)/He ages: Problems with He diffusion models or geological models?. Geochemistry, Geophysics, Geosystems, 20(5), pp.2418-2432.) describe this for AHe data. In many cases, it is unclear if sufficient temperatures have been reached to effectively remove all the previously accumulated "age" and any additional factors that may control age accumulation. It is not clear how noise is incorporated into the analysis. On line 236, are the dates of each individual measurement, for the case of AHe, shifted by a specified amount or are the uncertainties on the true age set based on the noise value. I think this is clarified on 249 where the ages are the correct age with an additional uncertainty. It would be interesting to know what happens when the input ages are drawn from a distribution given by the true age and a 10% uncertainty. But the 3% errors seem a bit small. Similarly, if a larger dataset of say 10 ages were measured, a larger spread in eU might be predicted and this would have important implications for the amount of information added by the AHe ages. Line Comments 63: mean etch figure width 68: I think you need remind people why AFT ages are ages and AHe are dates and not ages. My understanding is that date is preferred for AHe to reflect the idea that this does not correspond to a specific event. Surely this is equally true of AFT central ages? Why not just use age to be consistent with the AFT literature? 85-88: This sentence is a bit long. 256: It is not clear how the more complex models were rejected here. 285: "because the EX model undergoes a simple temperature weighting in QTQt,", this is not correct. The model integrates all parts of temperature weighted by posterior probability. 414: check that this is actually likelihood and not posterior probability. 415: greater uncertainty – this should really be greater certainty to mirror the idea that accuracy is closer to the true solution. 419: there isn't really an envelope of accepted models in a QTQt model. There will be lots of very bad models accepted

during the burn in phase for example, and in order to approximate the extremes of the posterior distribution, bad models need to be accepted. 442: Please be more specific about how noise is added. 500: I guess that the conclusions of Green and Duddy would probably be correct if the temps during the second burial peak event were a bit higher. 542: Has CRH been defined in this manuscript? It is probably worth describing what that is.

---

## Referee Comment (RC3) · Richard A. Ketcham (Referee) · 27 Dec 2020

Review of McDannell and Issler, Geochronology

In this contribution, the authors use synthetic data to demonstrate that a multi-kinetic approach to AFT (and AHe) data can be required for achieving a geologically meaningful thermal history. In their example case, neglect of the multi-kinetic nature of AFT annealing cannot be remedied by adding independent geological constraints, or additional data (AHe). The paper is put forward as a companion paper to an upcoming one in which samples with similar characteristics (very old, multi-kinetic) will be analyzed, and thus a certain amount of what's here may be book-keeping that will be useful for the next paper to refer back to. In general, I'm very supportive of this line of inquiry. The writing is very good, though it does get bogged down or venture off-topic occasionally. I think this paper is worth publishing and appropriate for this journal, but first some errors need to be fixed, and the focus needs to be improved.

This is a tricky kind of study to do, because it's difficult to generalize the problem – in which cases does multi-kinetics matter, and in which can it be neglected without overt penalty? For example, if there is fast cooling, subtle changes in kinetics will not matter too much. Put simply, careful attention to kinetics is likely to be most important in cases of long persistence at, or reheating to, a temperature range that differentiates the thermal responses of the grains present, and thus the ages and lengths recorded. It might be best to state this up front, and then pose the subsequent tests as a demonstration of that principle.

The synthetic data set is a bit over the top, in terms of quality. There are three kinetic populations, all equally represented in terms of grains and tracks. The synthetic t-T path has been designed to just touch into the lower part of the PAZ once for each of the two lower-resistance populations, in the necessary sequence for evidence of each to be preserved. More importantly, the authors say they add an "appropriate level of noise," but don't specify what that is, or how they did it. The age uncertainties on each population are all less than 5%, and all three populations have chi-squared probabilities of 100%, suggesting that the single-grain ages are under-dispersed compared to a true natural sample. This is borne out in the numbers in their Appendix; all grains but one are within 0.5-sigma of the central age; the one exception is almost exactly 1-sigma. This is massively under-dispersed compared to what one would expect with a natural sample (i.e., a random sample from a normal or Poissonian distribution), suggesting that something about how they generated these synthetic data was a bit off. This needs to be redone.

The result is a data set that grabs the viewer by the lapels and shouts "multi-population, multi-kinetic." Perhaps this was the point, but it does not make for effective advertising, as real data will never be this clear. It also doesn't make for a realistic test of the ability of thermal history inversion to read the history. They need to run the test with a normal degree of dispersion, and then it might be fun to run one with some excess dispersion, such as by adding some dispersion into the input kinetics, or adding another couple of small populations at different kinetics that are unidentifiable as populations because there are so few grains.

I think they could also have done more with testing along the lines of what's in Figure 5C, where they deleted a population. First, I disagree with the authors on that result – I think the penalty is surprisingly modest, to the extent of bringing up the question of how important that middle population is. If they had, say, a constraint for the depositional age of the initial sediments of the first burial episode, they could probably do without the second population entirely. This may be foreseeable if the lower part of the PAZ for the most-resistant population overlaps with the upper part of the PAZ of the least-resistant

one, thus providing effectively continuous coverage.  It also gets to a practical matter – if kinetics appear messy in the data, with lots of overlap (which there usually is), can the major information be extracted by concentrating on the end-members present?  This would be a useful question to answer, or at least address.  Second, another version of this exercise might allow the authors to run a sub-test – say, omit the highest-resistance population, see what they are missing, and then run the two lower-resistance populations as a single one, and see how that (also) messes up, in a 2-population case.  This would be a simpler case to reinforce the general point that reheating to the PAZ is when multi-kinetics gets important.

The authors lost me a bit in the discussion of rmr0; this section needs to be shortened and clarified, while keeping the important parts.  I have heard informally that people comparing the 2007 and 1999 models in some situations have noted divergent behavior, and have tended to prefer the 1999 one, but I have not seen a quantitative exploration of why this might be the case.  The authors claim that the difference may lie in the 2007 version weighting more resistant, and Cl-rich apatites versus apatites that feature cation substitutions.  This may be true, but I don't know; it's not clear that augmenting one aspect of compositional space necessarily diminishes the influence of another.  Alternative possibilities include other unknown or unexplored incompatibilities between the Carlson and Barbarand data sets, or that the empirical fitting method is oversimplified.  In particular, the 2007 model has a relatively compressed total thermal sensitivity range (maximum Tc = 180C in 1999 (note 2006 erratum), 160C in 2007) because of how the Carlson and Barbarand data sets interacted.  It's also worth being very clear that the kinetic meaning of rmr0 differs between annealing models – for example, Durango is 0.827 in the 1999 paper and 0.797 in 2007 (rounding down to 0.79 after applying rmr0+$\kappa$=1.04), but these values lead to the same closure temperature in their respective annealing equations.

The authors could also be a bit more clear when they discuss constraint boxes.  To the extent that imposed constraints embody reliable independent geological information, they are *always* proper to add.  Similarly, if the geology justifies them, there is no such thing as "excessively tiny" (line 473) – they should be the exact size the geology says they should be.  In fact, all paths that go outside those constraints contradict the geology, so why should they even be considered?  The two downsides the authors cite seem to some extent like red herrings.  First, there is the issue that QTQt seeks the simplest paths, which means it minimizes the number of t-T points.  By making a constraint box, they are basically telling one of those few points where it must be, reducing the freedom of the others, and giving the appearance of precision.  However, the broader credible interval given by an unconstrained QTQt is not a better reconstruction of the thermal history, because the parts of the envelope that overlap the otherwise neglected part of the true path are based on paths that violate the local geology.  (Naturally, the proper solution is to use HeFTy, which will broaden envelopes where the thermal history is poorly constrained by the data ;-) ).  Second, the authors caution about opening the door for assumptions [embodied in constraints] "to be heralded as geologic evidence".  The simple antidote is to formalize model reporting and document the reason for each constraint, as recommended by Flowers et al. (2015), reinforcing the idea that every constraint should have a reason for being there, and the modeler should be able to state what that reason is.  If the evidence underlying a constraint is uncertain, then models can be run with and without it, and the effects evaluated – simple!

The discussion features almost 2 pages on (U-Th)/He kinetics (line 514-574), which is really not the subject of the paper, and is not really further informed by any of the modeling work presented here.  It is basically editorializing, and most can be omitted.  It's unclear what the authors mean by the

"distorted" thermal history from using uniform-rmr0 RDAAM means that a non-FT-kinetics He model should be used (line 527-529).  An issue with the current alternatives (Gerin et al., 2017; Willett et al., 2017) is that they do not allow for variation of alpha recoil damage annealing kinetics at all.  If these kinetics vary, these other models would not be able to capture them, either.  More work needed…

It's not clear how the authors calculated their rmr0 value for Itambe apatite (line 137); I think it might be that they excluded Si from the "others" category.  This should be clarified.  It's not clear what they are trying to say in this part, and going back and forth between the two rmr0's from 1999 and 2007 is likely to be confusing unless things are very clearly set out.

The authors should make it clear that the 0.882 value of rmr0 for end-member OH-apatite (line 185) is an adaptation of fitted rmr0-$\kappa$ values of HS apatite (0.8559, 0.2206) to the simplification that rmr0 + $\kappa$ = 1, providing approximately the same closure temperature.  At least, I think that's what they did…

---

## Author Comment (AC1) · 21 Jan 2021

Below we address reviewers' comments. Reviewer's comments are in black text. Our replies are in blue text.

**Reviewer 1: Kerry Gallagher**

Perhaps the authors could try to reduce and reorganise the text to help the reader. For example, 2.5 pages on $r_{mr0}$ calibration is a bit of a distraction - much of that could go into appendix/supplementary (perhaps keep something on effective Cl for the main text).

We agree. This material is better suited for the next paper on using natural AFT samples to illustrate the multikinetic method. We will retain minimal information to explain what we did and the remaining $r_{mr0}$ discussion will be removed as it distracts from the flow of the paper.

There is a lot of detail from the forward model (predicted mean lengths, initial lengths) in the text that is also on the figures, so just keep the latter, or put all the numbers in a table (and the table perhaps in supplementary?).

We will make appropriate changes to reduce duplication.

Also, the results of using the wrong model (e.g., mono-compositional when it should be multi) are probably too long. I think most of us would appreciate that using the wrong model is likely to be a problem. The important point may be that we can still fit the data reasonably well (using a single sample).

We will try to reduce text where possible but we believe this is a very important point that deserves serious attention. We respectfully disagree with the last two comments. The data interpretation (data model if you like) is incorrect and therefore modelling produces a significantly different thermal history. We are not sure that most people can appreciate how different it can be without an example that illustrates this. The ability of the model algorithm to fit data closely is not a sufficient criterion for a good solution if the interpretation is erroneous. Generally, it is not difficult to fit data using a single AFT population. If there really is only one population, then it will provide useful constraints on the parts of the thermal history where it has sensitivity. However, for a multikinetic sample, separate populations can have significantly different thermal annealing behaviour. The single population interpretation means that all components are combined and treated as having the same annealing behaviour. This means that the length distribution, AFT age, and kinetics are different from the original synthetic data. Therefore, the ability of the combined data to discriminate different heating events is diminished because all grains are assumed to have the same thermal sensitivity by being lumped together. Inevitably, the model thermal history must change to accommodate this assumption. If multiple populations are unrecognized then the thermal history will be distorted in order to fit the data. The degree to which this happens is data-dependent. A very close fit to the data under these circumstances can give a false sense of confidence in the solution. Our single population example without constraints shows continuous cooling, completely missing the two separate heating and cooling events, and the geological interpretation of the results is quite different from the original history. If constraints are added, then thermal peaks appear but they are offset significantly in time and temperature with respect to the true model thermal history. Although the fit to the data is excellent for the combined sample, we do not consider this a reliable thermal history prediction. This is essentially the main point of the manuscript.

They need to state clearly up front the assumptions underlying some of the models - for example the multi-element/compositional models for the calibration to $r_{mr0}$ or effective Cl (eCl) are not perfect, are likely to contain correlations between the fitted parameters). If they do not have access to the original data or calibrations, then perhaps some kind of resampling could done (e.g. take some elemental composition data, resample those data using typical uncertainties and recalibrate the model).

We do not think that model recalibration is warranted or appropriate here and it is well beyond the scope of this short contribution. These comments underscore why we need to move the discussion of $r_{mr0}$ calibration to our next paper, which uses natural multikinetic samples, because it is peripheral to what we are trying to show here. First, these are synthetic data and they represent one example from an enormous range of

possibilities. We have already prescribed the kinetic parameters beforehand based on values that were determined for some natural samples in order to generate the synthetic data. This is not a real dataset so it is unclear what new insight will be gained by resampling and recalibrating the model. Model recalibration will not affect the two basic conclusions of this paper: (1) multikinetic data can contain a more detailed record of the thermal history than a single population, and (2) failure to recognize multikinetic behaviour could adversely influence thermal history results. You should arrive at similar conclusions no matter what kinetic model you choose. The $r_{mr0}$-based kinetic scheme of Ketcham is the standard for AFT modelling and the 1999 or 2007 model will lead to the same conclusions. Any new proposed kinetic models where annealing temperatures differ between populations will also lead to the same conclusions. Discussions of model calibration are more relevant for natural samples where real elemental data are transformed into kinetic parameters and this is a topic of our next paper.

Also, their example (synthetic data) are very clean and distinct in their compositions. Do we see/expect such well separated populations often, and if so how have these been dealt with previously? When does the ability to resolve the thermal history based on compositional groups start to deteriorate if the compositional groups are less distinct?

These are all important questions and they are best addressed by reference to a suite of natural multikinetic samples exhibiting different characteristics. A substantial number of Phanerozoic detrital samples from northern Canada show good population resolution but this cannot be demonstrated in the current paper. A future goal is to get these data released in a series of publications for different study areas. Our synthetic data are meant to be ideal to show that distinct populations can lead to well resolved thermal histories. To modify a phrase used by the reviewer, I think most of us would appreciate that the ability to resolve thermal histories will deteriorate as kinetic populations become less distinct. We do not see the point in investigating this for the current paper because the results will be unique to this synthetic example. There are many factors to consider (see above comments) that make generalization of these type of model results problematic. We think a more thorough analysis of factors governing model resolution should be informed by what is observed in natural samples.

Going to the extreme, the conclusion that we might draw from this study is that we should model each grain with its own specific compositionally defined annealing model (and model parameters). I agree with the authors that we often need to consider sub-populations of data from a given sample both for AFT and AHe, and that averaging the data prior to modelling is probably not a good idea (or at least we need to acknowledge that we will obtain some kind of average, perhaps unrepresentative, solution and that we are potentially throwing out information). However, the other side of the argument would propose that the predictive models are not that sophisticated, not free of uncertainty, and not even really based on a well developed understanding of the physical processes and how they operate on geological scales.

We do not agree that you can draw this extreme conclusion from the ideal synthetic data we are presenting in this paper. This sounds more like an expectation based on other experience. We think people have been preconditioned to expect poorly resolved AFT populations because proxy kinetic parameters that are in common use have low resolving power, and in addition, simply due to the classically low precision of the AFT method in general. Therefore, population overlap is the normal situation if multikinetic data are present in a sample, which is probably due to random geologic noise as well as imperfect kinetic model calibration. From our own experience, many natural multikinetic samples display distinct statistical populations when plotted using the elemental-based $r_{mr0}$ parameter and analysed using conventional approaches (age mixture modelling and radial plots). There are already published examples of natural multikinetic samples, so their existence in nature has been demonstrated. It is pointless to model single grains if, statistically, they fall into one of the discrete populations. The same criticism could be made of conventional modelling. Why not assign an individual Dpar value to every grain and model them that way. You would not do it because the modelling depends on the assumption that they are part of a single population. What we are doing is not so radical. Instead of assuming one overdispersed population, we are saying that elemental

variation can result in multiple populations being present and that exploiting this can result in improved t–T modeling results.

Again, none of this can be demonstrated in the current paper that deals only with one synthetic example. These topics are the subject of our next paper, which uses natural examples to illustrate multikinetic interpretation methods and modelling. We agree that empirical models have uncertainty and are a simplification of complicated underlying physical processes. However, it is unclear to us why a multikinetic approach to modelling would be more adversely affected than current modelling applications. It has been demonstrated clearly by laboratory annealing experiments that AFT annealing temperatures vary with changes in apatite composition (and this has been mostly dealt with unsystematically for decades). Why would ignoring this fact produce a better model?

Overall I think that main premise could be demonstrated more efficiently. The idea is that chemical composition (and perhaps associated mineral structure changes) has a major effect on annealing and diffusion in apatite, and this effect is multi-element, rather than just Cl/F as sometimes assumed for fission track annealing. It is a good idea to promote an analytical protocol of measuring a wide range of elements, rather than just Cl, or a proxy such as Dpar, as these data may be useful in future for annealing model recalibration and/or provenance (e.g. O'Sullivan et al. Earth Science Reviews 201, 2020). However, the available model calibrations are not based on a lot of data, and as stated in Carlson et al. (1999) "*in the absence of any physical understanding of why compositional variations impede or enhance annealing, we have little confidence that it can be used meaningfully to predict the annealing behavior of apatites not included in the experiments*". The concern is that these preliminary calibrations are assumed to give us the definitive model, free of uncertainty, and the rather strong conclusions about the inference of thermal histories are based on that assumption.

We agree that we need to focus this paper better to avoid distractions concerning model calibration, which is discussed in our next paper involving natural AFT samples. In this paper, the ability to convert real elemental data into kinetic parameters is not relevant. We have already predetermined that the synthetic sample has three multikinetic populations with different relative annealing behaviour based on what we have observed in natural samples. The question is, "What are the consequences for modelling the thermal history if you have populations with different annealing kinetics?" In principle, can more thermal history information be retained in a multikinetic sample than a monokinetic sample? It must be kept in mind that the quote from Carlson et al. (1999) is an expectation that the empirical model would not be reliable because of limited calibration. This is an inference that was not tested with follow up studies. Unfortunately, this statement may have deterred people from attempting to use the method. It is also worth noting that in the absence of a perfect understanding of *why* composition alters FT annealing behavior does not negate the fact that empirical evidence indicates it is a real phenomenon and does not ultimately prevent it from being useful.

For example, the results presented are based on more or less ideal data with well separated (kinetic/composition and age) populations (as described in 236 to 249). In this case, we can pretty much recover what we started with including heating out needing to specify near surface constraints on the thermal history. Furthermore, the modelling approach implemented in QTQt tends to prefer simple models, conditional on fitting the observed data adequately relative to more complex models. I think this should mean that individual models making up the credible interval range figure 3a will tend to look like the ML model shown in that figure, and we do not fully recover the thermal history of 20°C for about 500 m.y. prior to the first heating event ) - I would guess it is just the timing of reheating that changes, and probably the same for the second heating event.

Yes, using an ideal data set, we can recover much of the input thermal history except for the low temperature part where the model lacks sensitivity. Although this may appear to be a self-evident conclusion – modelling of multikinetic data is not a straightforward exercise. The ability to recover good solutions depends on the choice of modelling method and on how it is applied. Here, we demonstrate that good quality multikinetic data may preserve a record of multiple heating events under the right conditions.

We do not think that this is widely appreciated because people are used to dealing with single populations that are sensitive to a narrower range of the thermal history. This in part explains the fixation by many workers with whether or not AFT samples pass or fail the $X^2$ test — where passing does not absolutely ensure the absence of multiple sub-populations, nor does $X^2$ failure imply a 'poor quality' dataset or offer a direct means for understanding failure of the statistical test, which could occur due to many reasons such as high precision single-grain ages, large analytical sample size, or compositional heterogeneity (i.e., differential annealing). If none of these variables are assessed, dealt with, or ruled out – or the necessary data collected to do so – then how can the AFT community make any real progress towards advancing data interpretation and modeling practices?

The authors then demonstrate that combining these sub-populations and assuming an average composition (but generally fixed) leads to lack or resolution and/or spurious results for the inferred thermal history. The authors often imply that the latter is common practice, but do not really give any concrete examples. I think many, if not most, people working with fission track data are aware of the potential for over dispersed age data and hopefully would deal appropriately with an over dispersed population (using subpopulations based on composition or age, or perhaps remove egregious outliers - also this issue seems to have different significance depending on whether the data are collected with the traditional EDM method or LA-ICPMS, the latter method tends to have greater dispersion but similar central ages.

We think that modelling of mixed AFT populations is more common than realized but that it is unintended. We could cite specific examples but that could be viewed poorly, as we do not think the practice of 'multikinetic misidentification' is carried out on purpose, but instead with this paper and others in the future, we would hopefully draw attention to deeper investigation of complicated datasets by the AFT community. Awareness of overdispersed data and taking appropriate actions to understand it are two different things, with the latter likely yielding to the former in the majority of cases. If this were not so, there would be more examples to point to in the literature. We believe that multikinetic populations are best-resolved using elemental data (which are rarely collected) whereas population overlap is normal when using low-resolution kinetic parameters such as Dpar. If you cannot discriminate between populations, then modelling mixed populations is unavoidable. These issues cannot be addressed here with synthetic data but are the subject of the next paper. We will choose our words carefully here. We want to make the cautionary point that modelling multikinetic data as a single population can distort the results and it is something to take into consideration. In our experience, multikinetic populations are evident in both EDM and LA-ICP-MS AFT data. The reported higher dispersion with the LA-ICP-MS method is not a problem and both analytical approaches and derived data have been shown to provide similar results. The key is to have enough age and length data to characterize the different populations properly. This topic will be discussed in a subsequent paper that uses natural AFT samples.

The idea of AFT ages increasing then decreasing with Cl content was mentioned some time ago, I think by Barry Kohn, who had some data from Canada implying that the age decreases at high Cl (6% ??) which was associated with a change in crystallographic system from hexagonal to monoclinic. Not sure he ever published that though.

We are not sure what the reviewer wants here. We do not include real elemental data so this extra detail would not add significant information to our paper.

Not clear if the sensitivity of the data to composition as proposed is enhanced due to the long timescales, or this is a general feature.

This is a general feature of the data independent of timescale. We chose a deep time problem because it is harder to deal with than a Phanerozoic situation that may have more geological constraints available, and in general, deep-time problems are afflicted by greater uncertainty, where more data would be even more beneficial.

Also you switch between different combinations of AFT and AHe data, but the latter are fairly minor to the main proposition (and their sensitivity is based on the radiation damage model based on a FT annealing model). You could remove the AHe aspect totally and not change the message...expect perhaps in section 5.4, which comes pretty late and out of the blue.

We will try to rework the AHe part. It is true that the most important part of the paper concerns multikinetic FT data. However, we know that it can be difficult to reconcile AFT and AHe data sets for different reasons, especially for cratonic histories. We wanted to present one case where AFT composition may influence AHe modelling. Compositional data are not routinely used for AHe, but the commonly used radiation damage models are based on AFT annealing. If AFT annealing is truly affected by the composition then the radiation damage model should reflect the change in AFT annealing related to composition. We wanted to investigate this end member situation where all of the AHe age variation is controlled by radiation damage just to make a point that apatite composition may affect how alpha radiation damage accumulates. We will add a section upfront in the introduction to clearly outline why the AHe data are included and the reasoning behind the modeling. This should alleviate concerns that those data are a distraction, yet offer seeds for discussion for future work regarding if and how apatite composition affects He diffusion.

Lines 197 (and 213) There seems to be the assumption that AHe ages depend on composition, but is this just because the assumed model for radiation damage is one incorporating a fission track annealing model

The reviewer's statement is outside the scope of this paper but — yes, some published work has suggested composition may influence AHe ages directly but some work has suggested the opposite. No studies show a strong direct connection, but logic and the *ab initio* modeling suggest it should be so. The reviewer's point is a good one in that an additional issue could be secondhand via the kinetic model calibration. But annealing must involve diffusion, right? So the diffusion that heals tracks depends on small substitutions? The jury is still out. In our example, composition indirectly influences AHe ages because the radiation damage model is tied to AFT annealing, which can vary strongly as a function of composition.

Line 214 - QTQt will generate thermal histories regardless of ..data.... This is true of any sampling based modelling approach. The important point is how the generated thermal histories are accepted or not - QTQt effectively uses the ratio of data fit (likelihood) between a current and proposed model, while HeFTy, another piece of software for modelling thermal histories from thermochronological data, uses an absolute approach (p-value as a measure of data fit) for each thermal history. Perhaps accept is a better word than generate (also on 218), but keep the statement about the user needing to assess the output, particularly how well any particular model predicts the observed data.

Okay we will change generate to accept.

Line 249 - 3% seems small for AHe age data ??

We are assuming good quality, homogeneous AHe grains of suitable grain size and 3% is a normal analytical uncertainty without factoring in the Ft correction. We are trying to reproduce an observed date produced from a forward model, we are not modeling real data where data uncertainty may be cause for concern, when we in fact are trying to recover an unknown history. Ultimately the uncertainty should not be a major concern regardless of the value applied — especially in QTQt where better data quality is rewarded.

Line 256 - what happens if you do not constraint the heating/cooling rates ?

It is appropriate to determine suitable boundary conditions that are commensurate with the scale of the problem. The 5°C/Myr rate limit is more than ample for a timescale of 2 Ga. Given the unimodal and relatively broad length distributions and the old AFT ages, it is not possible to resolve extreme heating rates over this timescale. The main effect of not imposing a rate limit is to have longer model run times with the possibility of introducing rapid heating/cooling artifacts. This is especially deleterious for nondirected Monte Carlo schemes such as HeFTy and AFTINV. We will show results for the AFTINV model and discuss required boundary conditions that are needed to focus the model in promising areas of solution space. We

chose QTQt for modelling this deep time problem because it has a learning algorithm that refines solution space and model boundary conditions as it evolves. For nondirected Monte Carlo schemes, the model may generate millions of trial solutions and only converge on a small number of acceptable solutions if boundary conditions are too broad. Even with perfect data, it may be hard for nonlearning models to find answers over such large timescales.

Line 257 - "t-T points were only *accepted* if they provided.."..the points are added, but the important step is whether they were accepted or not.

We will change "added" to "accepted."

Line 262 - put the constraint definitions in section 3.1, and perhaps explain what they represent geologically

We can elaborate on the geological meaning of the constraint boxes. However, discussion of constraints does not belong in section 3.1 that deals with forward modelling (no constraint boxes are used here).

Line 264 - the ML model is potentially more complex than the MP model, but not always, and similarly the MP is not always simpler than the ML model...they can be the same.

We will reword to reflect this. The ML model is commonly more complex. The MP model is usually simpler.

Line 285 - not sure what you mean by a simple temperature weighting....the expected model is defined as

$$E(x) = \int xp(x)dx$$

and in this case the p(x) is the posterior. Given that the distribution of accepted models is the posterior then we just take the arithmetic mean of all accepted models to do that integral. The lower temperatures are because the distributions on temperature around the time of maximum temperature are often skewed (to lower temperatures) and so that leads to lower values for the expected (mean) maximum temperature. However, we often see that the duration of time close to the maximum temperature is greater for the expected model than say the ML model. This may tend to compensate a little in terms of fitting the data, but often not enough...you need to look at the predicted values relative to the observations. Note that we do not generally expect sharp V shaped thermal histories anyway (due to diffusion), but that is another issue.

This was a minor point and could be removed outright, but due to poor wording this should have more clearly stated that the averaging effect effectively pulls temperature downward. We will reword. The main point is the bias toward cooler temperatures.

Line 314 - QTQt does not use the central age directly (or even indirectly) as a model constraint. The predicted age for a given kinetic parameter and thermal history is used to infer the equivalent predicted $\rho s/\rho i$ ratio which is then used in the likelihood function with the measured Ns and Ni values for each grain (see Gallagher 1995, EPSL).

We will reword this statement. The input is the central age. How the model uses that information is another issue. We are trying to avoid going into too many fine details of the QTQt modeling because this information has been published elsewhere.

Section 4.2 -line 300 perhaps just discuss using the correct kinetics (here you choose rmr0, but this could be another parameter, even if less sensitive ?)

The point is that the kinetics are known in advance for this synthetic data set. The Ketcham et al. (1999, 2007) annealing model uses $r_{mr0}$ values. If users specify Cl content or Dpar, they are converted to $r_{mr0}$ values using empirical correlations for the purpose of the kinetic model. We chose $r_{mr0}$ based on our experience that elemental-derived $r_{mr0}$ values provide better kinetic population resolution than Dpar or Cl content alone. We cannot show that in this paper because we do not include real data for a natural sample.

Line 317 - when using the average eCl 0.213±0.373 in QTQt, this implies that you let the kinetic parameter vary as part of the inevrsion...what was the distribution of the accepted values ? If skewed to a higher or lower value, that may be informative concerning the sensitivity of the different subgroups of data to an average/common kinetic parameter.

We can show this information in the supplement if necessary. Yes, the kinetic parameter was allowed to vary during the inversion, however regardless of skew in either direction, the point is that you could assign any kinetic value here within reason, *but more importantly*, if a multikinetic sample is misinterpreted as a single population and the true history is complex (like we see in our example), then the fit to the observed data may be perfect, but one would never recover a history that closely resembles the true history. Therefore, as the reviewer mentioned previously the model assumption would be incorrect, but in this case both the t–T model AND the assumed single population are both incorrect. There seems to be a misperception in the reviewer's statement because the sensitivity of the different subgroups are lost entirely when combined into a single, monokinetic population — hence the point of this exercise. Combining the data yields some 'average' AFT age and some average kinetic value that removes t–T sensitivity. It is worth noting that ***not all AFT samples are multikinetic*** and we are not claiming this. Each natural sample is unique, and some may be multikinetic and others may not be, but if compositional data are not collected in the first place, how can this be known or addressed?

Line 318 - perhaps some examples of overdispersed data treated as a single population. The impact of this is likely to depend on how overdispersed and why...failing the chi-sq pvale = 5% test is not necessarily the definitive indication (e.g. we can pass at a level of 5.0001, but fail at 4.9999). Leaving aside analytical problems, dispersion that is a real if sometimes unwelcome signal could be due to compositional effects and discrete provenance related age populations (for which compositional ranges may be similar).

We will reduce some of this section. Based on the reviews, we think part of this discussion is best left to the next paper that includes real data for natural samples. Without including real data, we cannot show why we think unresolved, mixed AFT populations may be a larger issue than recognized. However, we do provide an example of overdispersed data being treated as a single population in our model examples. See our comment above regarding the $X^2$ test.

Line 320 - we do not necessarily have to formally identify discrete groups with mixure modelling, but just divide compositional range into subgroups and use the appropriate values (e.g. as Geotrack seem to do for Cl binned at 0.1wt % intervals)

Yes, but how many people do this other than Geotrack? That isn't clear in the literature overall and it isn't very clear at all how Geotrack carries out thermal history analysis. We think discrete models work better than a more continuous model for the natural multikinetic samples we have observed. However, we cannot demonstrate this here with a synthetic example, so it is beyond the scope of this paper. Our synthetic example is based on features we have seen in natural samples. Dealing with the distribution and allocation of track lengths in either a discrete or continuous approach is the most important factor to consider, but is outside the scope here.

Line 330 - it is not that QTQt failed to reproduce the true AHe dates...it is because the wrong choice of model prevented QTQt from doing so....

We will reword to explain that the true AHe dates were unable to be reproduced due to incorrect data treatment, i.e., incorrect model choice (see 'line 317' comment above as well)

Line 359 - I think that Geotrack do use compositionally discrete modelling for their routine AFT studies, but we rarely get to see the predictions for their preferred models.

That may well be but unless it is clearly documented in the literature then we are not privy to the details and can only speculate on how things were done. We still think it is fair to say this is underutilized in the scientific literature.

Line 388 - data quality is important too...

Sure, we did not list everything. Data quality is a factor but it is a function of many things. You need enough measurements to define populations, you need representative measurements for each grain and this could be affected by zoning, stressed grains with dislocations, problems with analytical procedures, etc.

Line 391 - what do you mean by more extensive ?

Greater time-temperature ranges. We can state this more clearly.

Line 396 - as I said above, perhaps we should model each grain with a specific set of kinetics ? This does no necessarily require running N annealing models for N grains, but perhaps 4-5 and we can interpolate the results (e.g. predicted ages and length distributions) for intermediate compositions.

We think the discrete model is a better way to go. You have well defined discrete populations. Why pick something in the middle and subdivide these well-defined populations? What is to be gained by further subdivision? If coherent age and length populations appear on plots of AFT parameters versus kinetic parameter, why resolve populations to a finer scale than necessary? In any case, this is a synthetic sample with predetermined properties. We also stress that arbitrary division into populations may be problematic depending on how this is carried out. With real samples we use radial plot mixture modeling to first identify discrete populations and in nearly all cases these populations coincidentally or naturally align with breaks or divisions within the AFT age data with respect to apatite composition. This agreement suggests to us that there is real kinetic/differential annealing giving rise to variability in single-grain ages and that those ages in turn correlate with changes in composition. After all, if single-grain dates correlate with composition and the t–T path of the sample caused differential annealing, we should expect a $X^2$ test failure and mixture models should reveal discrete subgroups.

Line 401 - without necessarily requiring

We can add this qualification.

Line 407 - what was the first hand ? I had forgotten by the time I got here.

We will tighten up the text so you remember.

Line 413-415...not sure I understand this...explicit condition on the model prior ? Perhaps you mean on the model sampling.

Yes, will reword.

Line 415 - not correct...it is the posterior that will be lower I think, rather than the likelihood. At least the acceptance of models is based on the posterior

We will rephrase this to be correct in our Bayesian language

Line 418 assist - use focus perhaps.

We will change this.

Line 421 - proper data interpretation is vague...you mean some of sub-population classification based on composition, age dispersion ?

We will replace proper data interpretation with kinetic population interpretation.

The discussion of precision v accuracy based on the credible intervals does not really take account of the form of the individual thermal histories...although they may define an envelope that seems consistent with the known true answer, the actual thermal histories may not really capture the true thermal history as well.

Yes, we know that QTQt calculations differ from models such as HeFTy and AFTINV where the plotted individual solutions are required to fit the data at a specified level of significance. Depending on how plots look we may add all individual paths to show how and where the Expected ± 95% envelope falls.

Line 429 - again, it is difficult to assess if we really resolve the true thermal history from the credible intervals..they do not tell us about the variation of individual thermal histories...so the earlier reheating event may start at different times, but we do not necessarily resolve the period sitting at low temperatures for a long time.

Yes, we know the issue with the individual solutions. The important point is the main features of the forward model thermal history are recovered by the Maximum Likelihood and Expected models. See point above.

Line 434 - more complex thermal histories were accepted or retained rather than added.

We will change added to accepted.

The rest of this paragraph to line 439...when you allow more complex models that do not improve the data fit to be accepted, you start to sample the prior more extensively, i.e. you tend to fill up the prior box in those parts of the time-temperature space where the data do not care what happens. For example, prior to a reheating event, the temperature can be pretty much anywhere from the reheating maximum value to the lower limit of the temperature prior. This does not mean we resolve the thermal history there. Almost the opposite. All we resolve is that the temperature has to be lower than the subsequent maximum.

Makes sense. The model lacks sensitivity at low temperature and it is allowed to fill up the space. We will modify the text.

Line 443 - Apfu Cl ?..if you add normally distributed noise, or even uniformly distributed, with the same mean value as the true model, it is not a great surprise that the expected model and credible intervals do not change much as we will tend to fit the average values of the data (which will reflect average values of the kinetics, which will be sampled on average to give the correct value...if that makes sense).

We will add eCl after apfu in the brackets. Kinetic parameters are relative. If the mean is similar to the "true" kinetic value used in the forward model, the inverse model will converge to be close to the input values. If mean values are not close to the input value, the model may converge on a different value but it will need to offset the remaining kinetic parameter to satisfy the observations. The result is that model temperatures can shift upward or downward but still converge to the same relative temperature history. Depending on how input ranges are specified, final kinetic parameters can be anywhere within the input range. The model will have trouble converging on good solutions if kinetic parameter ranges are too narrow to accommodate the required adjustment to kinetic parameters. None of this changes the conclusion of this paper that multikinetic data may contain a record of multiple heating events.

Line 448 - this acknowledgement of imperfect kinetic models should be stated much earlier, and ideally quantified somehow..(i.e. the recalibration exercise I mentioned above).

Recalibration is not important for this paper because we are using synthetic data. We assume the kinetic parameters are known and we investigate whether we can recover important thermal history information from a multikinetic sample using inverse modelling. This drives home the point that our current kinetic models do have some validity. All AFT modelling suffers from the same uncertainty on kinetic parameters. The most important aspect of a multikinetic model is the relative behaviour of the system. Calibration affects the absolute model temperatures. Relative kinetic behaviour preserves important details of the thermal history and this is important distinction will be discussed in the next paper.

Line 456-58 - the data quality (and the associated information on the thermal history) is first order in terms of what we can resolve. If the data are poor, scarce, we want more uncertainty.

Yes, data quality is critical. Uncertainty increases as data quality degrades. There are many factors influencing model results. These should be explored in future papers after as we learn more about natural multikinetic samples.

Section 5.3 - this deals with a separate issue to the main thrust of the paper, and adds to the feeling of stream of consciousness. I think it is not just with QTQt that we need to assess the effect of constraints on the final results and this has been said before in the exchanges in Earth Science Reviews over the last few years, and still ongoing. Any model result is conditional on the assumptions made to obtain the result. Thus a no constraint model can often be relatively boring (i.e. linearish cooling) but if the data can be adequately explained that way, then it is useful to know and can be considered as an end member model. Adding constraints is not a problem, but these need to be justified and this needs to be clearly stated in any study, preferably with some assessment of the confidence in a constraint. Additionally, when a thermal history is composed of linear segments joining up constraint boxes and we fit the data, this does not mean the constraints are justified...just that the data do not contradict these constraints and they do not require more complexity than imposed by the constraints.

A combination of inverse models and targeted forward models, often based on the inversion results, can be useful to deal with many of the problems discussed in this section and you do say that ..but you could just say that more concisely.

Yes, we will streamline this. There are many different situations and it is hard to generalize based on modelling one synthetic data set. There are different modelling strategies. Nondirected Monte Carlo models such as HeFTy and AFTINV usually require constraints in order to restrict model search space or they may have great difficulty converging. We just wanted to point out that constraint boxes might not be helpful under certain conditions. If multikinetic behaviour is unrecognized due to a lack of compositional data, for example, then thermal histories may be distorted to fit the data (see above). Use of constraint boxes will not lead to a better solution under these conditions.

Lines 474-476...not clear what this means...especially linearizing bias and Bayesian treatment of user constraints ..?

We will remove this text as it is not entirely necessary and as we can see from the comment that it can be confusing or misleading. What we were attempting to point out is that if 'more complex' models are prevented from being accepted (regardless of likelihood) then if there are two boxes placed within QTQt, the tendency will be to connect the two boxes with a simple 'linear' history segment, **IF** the data do not require a more complex path to provide a better fit. This does not mean that linear segment is legitimate or 'real' — which is where the point regarding a basement nonconformity being subaerial 100s of Myr before the geologic evidence (and a small constraint box) would suggest. This is likely an artifact of thermal history construction. Without boxes, the model fills the low sensitivity, low temperature space. Surface temperature can occur anywhere in the interval. When all solutions are forced to be at surface early on then temperatures ramp up because of the way temperature histories are constructed. The input history is very difficult to fit at low temperature because it is hard to have an effective heating rate of zero persisting over such a long time interval unless you force the model to do so. You can fit the data without this requirement.

Line 494..I agree with the sentiment...heralded perhaps better stated as imposed under the guise of geological evidence. This goes back to what I said above - there is not a problem using constraints/forcing a model, but the results are conditional on these and there are generally other models that can fit the data without, or with different constraints. Again, it is how valid the constraints are that is the big question.

Yes, we agree constraints have their place. We will rework the section to make things clearer.

Line 504...again Geotrack seem to use 0.1 Cl wt% bins when modelling, but we never see their predictions. You may add here that limitations arise when modelling samples independently, especially in boreholes. It is clear that jointly modelling multiple samples (plus the mutli-kinetic approach for each sample) is better -

data noise will tend to cancel out (if it is random) while real signal (the thermal history) should be reinforced. See Gallagher et al EPSL 2005 for an example with synthetic data too...

We expect models that use 0.1 wt% Cl bins will give different results than those that model discrete multikinetic populations. In the former case, kinetic parameters are expected to vary smoothly. We do not see this in most of our multikinetic samples. Instead, there are significant differences in kinetic behaviour between discrete populations. We also do not think a single parameter like wt% Cl will properly account multikinetic behaviour. This topic cannot be discussed in the current paper. It needs to be mentioned in the context of naturally occurring multikinetic populations that is the subject of a future paper.

Liner 509-510...what if the true thermal history is simple cooling ?

If that is the situation then inverse multikinetic modelling should yield linear cooling histories. We have unpublished natural examples where simple cooling can explain observed multikinetic data. In other cases, complicated histories are required (consistent with the geological complexity of the regions from which the samples were obtained).

Line 513- not sure what you mean by universal slow cooling suppositions...there is nothing forcing cooling in QTQt, apart from perhaps have a sample at surface temperature today and letting it start hotter...if the data are happy with that, then why not ? If the data need more complexity the thermal history should adapt.

This may be true for QTQt. However, other models such as HeFTy and AFTINV can be used to enforce continuous cooling, or enforce anything for that matter. These models may fit at the threshold of acceptability (and therefore cannot be rejected as a possibility) but may not be able to fit the data closely. The question is, "should we prefer the simple model that barely passes and has trouble finding solutions or a somewhat more complicated model that fits the data closely and converges quickly?" Complexity often yields better fits to observed data. As sample quality increases, it usually becomes easier to answer this question. With enough data, the linear cooling model may not reach the acceptance threshold. In any case, treating multikinetic data as a single population may lead to a significantly different and more simplified thermal history. The comment about *"universal slow cooling…then why not?"* highlights the issue cropping up in the literature with the use of QTQt as a black box where people dump in lots of data and hit "go" and especially over long timescales, these data can often be adequately reproduced under linear cooling assumptions. This doesn't mean linear cooling is valid. This is a difficult problem to address.

Much of section 5.4 is relatively speculative but more importantly deviates from the main message of the paper concerning FT annealing and left me a little confused. The last paragraph is OK...but I think you could drop much of this and perhaps save it for another paper....but that is up to the authors. While I agree that allow an effective parameter to vary in the modelling, as QTQt allows, can be useful - the paper by Ricanati et al. demonstrated that yes we can improve the fit to the data by having a wiggle factor for each grain diffusivity. As Samuel Karlin said, 'The purpose of models is not to fit the data but to sharpen the question'...so demonstrating we can fit the data with an additional factor is not really the solution, but suggests we should look for a physical control on that factor. Given the arguments here, I would say go and measure apatite chemistry to demonstrate that there is some correlation of age and chemistry or effective diffusivity then you are on to something (perhaps this will be possible with the method, Pickering et al. mentioned). For me, as it is, the sceptic will take this section as special pleading for a control on AHe date dispersion that is neither understood nor constrained. Those adopting the averaging strategy for AHe data will just say we do not understand enough to do anything more sophisticated and carry on. I would add a caveat to the Karlin quote too....not fitting the data means we need to ask different questions.

We agree that we need to scale this back and save it for the next paper where we can have more in depth discussions around multikinetic interpretation and modelling. We also agree that we are only showing one possible reason why AHe data may be difficult to model and that much more work is needed to resolve the problem. We will focus the paper more on the key points we want to convey. We would also add that if the thermochronology community does not adequately understand the problems surrounding AHe data and

workers have to manipulate or massage dates through averaging, 'culling outliers', or 'eU binning' should papers be published utilizing these data or should more work be done to address these issues head on? This aligns with the reviewer's sentiments or caveat to the Karlin quote.

Line 582...Steve Bergmann was often insistent on the importance of OH as a control on fission track data....not sure if he ever published anything though.

We think this is well established in the literature and that OH is an important factor and it should be included when obtaining detailed elemental data for constraining kinetic parameters. That is why electron microprobe analysis is preferred over LAICPMS elemental data. Carlson et al. (1999) and Ketcham et al. (1999) noted that it was important enough to be a separate term in their empirical $r_{mr0}$ equation. They also noted that $r_{mr0}$ correlated better with OH than Cl or Dpar. This topic is peripheral and better discussed elsewhere.

Line 594 - what recent publications ?

The section on AHe will either be removed entirely or greatly reduced and reworked so these points may no longer be relevant. Also the AHe data will be downplayed in terms of significance and discussion. The relevant citations being mentioned in the previous section being Gautheron et al. 2013 or Gerin et al. 2017, Recanati et al 2017 and Powell et al. 2017 and 2020 for looking at both AFT and AHe composition/kinetic variability together.

Line 595 - not misfit, but poor fits.

We will change this.

Line 596 - degrades in what sense ?

This is a sample specific result. It is hard to generalize. For our case, it results in a simpler thermal history or a change in the timing and magnitude of thermal events relative to the input history (if constraints are enforced).

It may be the paper may be better concluded by adding a series of recommendations on analytical practice/protocol and then modelling strategies.

We cannot say too much here because we are not providing a detailed description of how to interpret multikinetic data for natural samples. Much of this will be dealt with in a future paper.

 Fig 1 - perhaps put the 2 constraint boxes on the forward thermal history. They are a little strange as constraints as we might expect the constraint to be the stratigraphic age of sediment deposited at the time of the start of the heating events, rather than some time prior to the heating event.

The constraint boxes do not belong on this figure. They were not used to generate the forward model. They are considered as something inferred from fragmentary geological evidence. We can elaborate on what they represent geologically.

Fig 3 - the credible interval ranges are fine as presented in this figure, but it may be useful to put the sampling of individual models and/or the marginal distribution (the coloured plot from QTQt for a given sample thermal history) in the supplementary - then we can see how many models actually start reheating at 1200 Ma for example.

We will add more QTQt output to the supplement to assess overall model behavior.

---

## Author Comment (AC2) · 21 Jan 2021

**Response to Reviewer Comments**

We would like to thank all reviewers for their comments. There is some overlap concerning the issues being raised by the reviewers and this is most helpful for identifying where the manuscript needs changes, particularly in the introduction where we need to state more clearly the rationale and assumptions behind these tests. We agree that the manuscript can be shortened in places and that some material (e.g., $r_{mr0}$ discussion) is better suited for a second more comprehensive manuscript we are working on that uses natural samples to illustrate multikinetic AFT data interpretation and modelling techniques. We plan to rework and shorten the discussion and introduction of this manuscript and make it more focused on the key objectives and conclusions. Discussion of some of the broader implications of the multikinetic approach are better suited for our other manuscript dealing with natural samples. Multikinetic AFT thermochronology is not an easy subject to present and therefore we need to include enough detail in our reply to provide context for what we are doing and what we hope to accomplish. Unfortunately, a large body of relevant multikinetic data remains to be published and we are trying to release it in a logical and orderly fashion.

The reviewers raise valid points concerning the ability to resolve thermal histories with overdispersed kinetic populations. We agree that there are multiple factors that influence the quality of thermal history reconstructions, which were briefly mentioned in the manuscript. However, this relatively short contribution was not intended to address these issues. The main point of this paper is to show that true multikinetic AFT samples with significantly different annealing characteristics carry far more thermal history information than single AFT populations with typical annealing temperatures (~110°C). We may have diluted this point with too much discussion of tangential phenomena, but that will be resolved in our revision. Although this may seem obvious, we think the amount of detail that can be obtained from a high quality multikinetic sample is underappreciated and that examples of clear multikinetic behaviour are grossly underrepresented in the published literature. Reasons for this are discussed in another paper we are working on that deals with natural samples that will also likely be submitted to GChron as a companion to this paper.

We think that it is best to use an ideal endmember synthetic sample with well-defined kinetic populations to illustrate that it is possible, in principle, to recover information about multiple heating events from a single multikinetic AFT sample using inverse modelling techniques. One can always add more noise to the data until you degrade the sample to the point where thermal histories are resolved poorly. This approach can be misleading because you are only examining a narrow range of possibilities. There are an infinite number of scenarios when dealing with real multikinetic data and this makes it very difficult to generalize modelling simulations using synthetic data alone. This is why we used the 'perfect' synthetic data examples to clearly show that multikinetic interpretation works well within the current AFT kinetic framework, regardless of the known limitations of that framework. This could be viewed as circular, but our experience with real datasets proves to us that this is not the case. The conundrum faced is, do we show complex/messy natural AFT data and try to prove that multikinetic interpretation is valid and worthwhile (while also dealing with geologic uncertainty and various other complications) — or do we use synthetic data to merely demonstrate that in the ideal case multikinetic interpretation works and provides beneficial information. The former is muddied by unexplained complexity that potentially blur data trends, whereas the latter is a narrow sampling of a specific scenario from many different possibilities.

We do not think that any reasonable thermochronologist would argue that 'multikinetic AFT' is not a real phenomenon, given what we know regarding the empirical relationship established between track annealing and apatite composition, nor would they argue that more data is somehow not beneficial. We think these types of sensitivity analyses are important, but it is premature at this stage to do this until more examples of natural multikinetic AFT data are published to help guide the modelling. We do not see the value in degrading synthetic data quality to be more in line with what people expect for real data because then questions would arise surrounding data interpretation, which is better left for natural samples. If the reverse were true and we presented data from 'messy' natural samples where data may be distributed unevenly between kinetic populations or populations may be resolved poorly – then there is the possibility of criticism about whether or not a multikinetic approach is valid or how it was implemented. Therefore,

showing that multikinetic AFT works with perfect data implies that any encountered complexity with natural samples is due to unforeseen geologic variables and the nuances of AFT analysis (i.e., grain selection, sample quality, grain size, isotopic zoning, etc.).

Analytical uncertainty is only one of a host of factors that influence our ability to recognize and successfully model multikinetic AFT populations. Based on a large suite of unpublished Phanerozoic detrital samples from across northern Canada, we believe that the natural geological variability in samples is a more important factor than analytical uncertainty. Even with perfect measurements, the ability to resolve different kinetic populations depends on the number of AFT age and length measurements, and their distribution across different populations. Multikinetic AFT data with low U apatite grains can pass the $X^2$ test due to large uncertainties on single-grain ages and these can be misinterpreted as single populations if not carefully investigated using elemental data. If compositional zoning is present, some apatite grains can be assigned to the wrong kinetic population. Some populations may be too track retentive or have too low retentivity to be sensitive to key parts of a thermal history. In addition, the ability to retain a record of a complicated thermal history depends strongly on the relative timing and magnitudes of different thermal events and this in turn feeds back into whether kinetic populations have experienced enough differential annealing to be clearly resolved. In spite of these and other factors, we have observed that many highly overdispersed detrital samples from northern Canada have well resolved kinetic populations with low dispersion (0-10%) and high $X^2$ probabilities (30-100%) when defined using detailed elemental data. Until more AFT studies that include elemental data are published, it is difficult to make assumptions about errors and the ability to resolve kinetic populations.

Our synthetic sample is ideal in the sense that the age dispersion is low for individual populations (but it is high if the combined data are treated as a single population) and the thermal history has a sequence of heating events of decreasing magnitude. This style of multiple heating events of decreasing intensity is quite common in northern Canada and it has been inferred for Precambrian Shield rocks based on AFT modelling. Despite appearances, the kinetic parameters have not been optimized to produce the best thermal history results but instead represent populations that we have encountered in natural samples and were previously discussed in the AFT literature. If we wanted to choose kinetic parameters that yield the best results for the thermal history, we would have narrowed the kinetic parameter range so that populations 1 and 3 would be more sensitive to the two heating events. Population 2 has an eCl or $r_{mr0}$ value that is in the typical range for apatite whereas population 3 has extremely high track retentivity and population 1 has low retentivity. Therefore, we do not think our scenario is a rare situation that has been contrived to make things work.

It is true that dispersion is very low for the individual kinetic populations, but we wanted to determine whether inverse modelling could converge on a reasonable solution using high quality synthetic data. Our example represents the more difficult "deep time" problem where geological constraints are few and the thermal history spans a 2 Ga time range. Although this modelling may appear to be straightforward, it is not. We chose QTQt for modelling because it incorporates a learning algorithm that does a better job of exploring solution space in the absence of geological constraints. Programs such as HeFTy (Ketcham, 2005) and AFTINV (Issler, 1996) that use nondirected Monte Carlo schemes have great difficulty in converging on solutions unless constraints are imposed to limit model search space regardless of how ideal the data are. In order to illustrate this, we will include AFTINV model results for the synthetic sample and discuss the constraints that were needed to obtain solutions.

In summary, this contribution is not meant to illustrate how to do multikinetic interpretation and modelling but instead to show that complicated thermal history information may be retained in multikinetic AFT samples. For natural samples, the degree to which this information can be recovered will vary from sample to sample. This paper is not intended to explore how model resolution varies with data dispersion. We believe this type of modelling should be informed by what is observed in natural samples. In a subsequent paper, we plan to demonstrate that multiple thermal events are necessary to fit AFT parameters for natural samples with multikinetic populations. In order to show that information on multiple heating events can be

retained in natural multikinetic samples, we need to demonstrate this possibility using synthetic data. Furthermore, we want to show that model results can be distorted if multikinetic behaviour is not recognized. We will revise the manuscript to more accurately reflect our intent.

Below we address reviewers' comments. Reviewer's comments are in black text. Our replies are in blue text.

**Reviewer 1: Kerry Gallagher**

Perhaps the authors could try to reduce and reorganise the text to help the reader. For example, 2.5 pages on $r_{mr0}$ calibration is a bit of a distraction - much of that could go into appendix/supplementary (perhaps keep something on effective Cl for the main text).

We agree. This material is better suited for the next paper on using natural AFT samples to illustrate the multikinetic method. We will retain minimal information to explain what we did and the remaining $r_{mr0}$ discussion will be removed as it distracts from the flow of the paper.

There is a lot of detail from the forward model (predicted mean lengths, initial lengths) in the text that is also on the figures, so just keep the latter, or put all the numbers in a table (and the table perhaps in supplementary?).

We will make appropriate changes to reduce duplication.

Also, the results of using the wrong model (e.g., mono-compositional when it should be multi) are probably too long. I think most of us would appreciate that using the wrong model is likely to be a problem. The important point may be that we can still fit the data reasonably well (using a single sample).

We will try to reduce text where possible but we believe this is a very important point that deserves serious attention. We respectfully disagree with the last two comments. The data interpretation (data model if you like) is incorrect and therefore modelling produces a significantly different thermal history. We are not sure that most people can appreciate how different it can be without an example that illustrates this. The ability of the model algorithm to fit data closely is not a sufficient criterion for a good solution if the interpretation is erroneous. Generally, it is not difficult to fit data using a single AFT population. If there really is only one population, then it will provide useful constraints on the parts of the thermal history where it has sensitivity. However, for a multikinetic sample, separate populations can have significantly different thermal annealing behaviour. The single population interpretation means that all components are combined and treated as having the same annealing behaviour. This means that the length distribution, AFT age, and kinetics are different from the original synthetic data. Therefore, the ability of the combined data to discriminate different heating events is diminished because all grains are assumed to have the same thermal sensitivity by being lumped together. Inevitably, the model thermal history must change to accommodate this assumption. If multiple populations are unrecognized then the thermal history will be distorted in order to fit the data. The degree to which this happens is data-dependent. A very close fit to the data under these circumstances can give a false sense of confidence in the solution. Our single population example without constraints shows continuous cooling, completely missing the two separate heating and cooling events, and the geological interpretation of the results is quite different from the original history. If constraints are added, then thermal peaks appear but they are offset significantly in time and temperature with respect to the true model thermal history. Although the fit to the data is excellent for the combined sample, we do not consider this a reliable thermal history prediction. This is essentially the main point of the manuscript.

They need to state clearly up front the assumptions underlying some of the models - for example the multi-element/compositional models for the calibration to $r_{mr0}$ or effective Cl (eCl) are not perfect, are likely to contain correlations between the fitted parameters). If they do not have access to the original data or calibrations, then perhaps some kind of resampling could done (e.g. take some elemental composition data, resample those data using typical uncertainties and recalibrate the model).

We do not think that model recalibration is warranted or appropriate here and it is well beyond the scope of this short contribution. These comments underscore why we need to move the discussion of $r_{mr0}$ calibration to our next paper, which uses natural multikinetic samples, because it is peripheral to what we are trying to show here. First, these are synthetic data and they represent one example from an enormous range of possibilities. We have already prescribed the kinetic parameters beforehand based on values that were determined for some natural samples in order to generate the synthetic data. This is not a real dataset so it is unclear what new insight will be gained by resampling and recalibrating the model. Model recalibration will not affect the two basic conclusions of this paper: (1) multikinetic data can contain a more detailed record of the thermal history than a single population, and (2) failure to recognize multikinetic behaviour could adversely influence thermal history results. You should arrive at similar conclusions no matter what kinetic model you choose. The $r_{mr0}$-based kinetic scheme of Ketcham is the standard for AFT modelling and the 1999 or 2007 model will lead to the same conclusions. Any new proposed kinetic models where annealing temperatures differ between populations will also lead to the same conclusions. Discussions of model calibration are more relevant for natural samples where real elemental data are transformed into kinetic parameters and this is a topic of our next paper.

Also, their example (synthetic data) are very clean and distinct in their compositions. Do we see/expect such well separated populations often, and if so how have these been dealt with previously? When does the ability to resolve the thermal history based on compositional groups start to deteriorate if the compositional groups are less distinct?

These are all important questions and they are best addressed by reference to a suite of natural multikinetic samples exhibiting different characteristics. A substantial number of Phanerozoic detrital samples from northern Canada show good population resolution but this cannot be demonstrated in the current paper. A future goal is to get these data released in a series of publications for different study areas. Our synthetic data are meant to be ideal to show that distinct populations can lead to well resolved thermal histories. To modify a phrase used by the reviewer, I think most of us would appreciate that the ability to resolve thermal histories will deteriorate as kinetic populations become less distinct. We do not see the point in investigating this for the current paper because the results will be unique to this synthetic example. There are many factors to consider (see above comments) that make generalization of these type of model results problematic. We think a more thorough analysis of factors governing model resolution should be informed by what is observed in natural samples.

Going to the extreme, the conclusion that we might draw from this study is that we should model each grain with its own specific compositionally defined annealing model (and model parameters). I agree with the authors that we often need to consider sub-populations of data from a given sample both for AFT and AHe, and that averaging the data prior to modelling is probably not a good idea (or at least we need to acknowledge that we will obtain some kind of average, perhaps unrepresentative, solution and that we are potentially throwing out information). However, the other side of the argument would propose that the predictive models are not that sophisticated, not free of uncertainty, and not even really based on a well developed understanding of the physical processes and how they operate on geological scales.

We do not agree that you can draw this extreme conclusion from the ideal synthetic data we are presenting in this paper. This sounds more like an expectation based on other experience. We think people have been preconditioned to expect poorly resolved AFT populations because proxy kinetic parameters that are in common use have low resolving power, and in addition, simply due to the classically low precision of the AFT method in general. Therefore, population overlap is the normal situation if multikinetic data are present in a sample, which is probably due to random geologic noise as well as imperfect kinetic model calibration.

From our own experience, many natural multikinetic samples display distinct statistical populations when plotted using the elemental-based $r_{mr0}$ parameter and analysed using conventional approaches (age mixture modelling and radial plots). There are already published examples of natural multikinetic samples, so their existence in nature has been demonstrated. It is pointless to model single grains if, statistically, they fall into one of the discrete populations. The same criticism could be made of conventional modelling. Why not assign an individual Dpar value to every grain and model them that way. You would not do it because the modelling depends on the assumption that they are part of a single population. What we are doing is not so radical. Instead of assuming one overdispersed population, we are saying that elemental variation can result in multiple populations being present and that exploiting this can result in improved t–T modeling results.

Again, none of this can be demonstrated in the current paper that deals only with one synthetic example. These topics are the subject of our next paper, which uses natural examples to illustrate multikinetic interpretation methods and modelling. We agree that empirical models have uncertainty and are a simplification of complicated underlying physical processes. However, it is unclear to us why a multikinetic approach to modelling would be more adversely affected than current modelling applications. It has been demonstrated clearly by laboratory annealing experiments that AFT annealing temperatures vary with changes in apatite composition (and this has been mostly dealt with unsystematically for decades). Why would ignoring this fact produce a better model?

Overall I think that main premise could be demonstrated more efficiently. The idea is that chemical composition (and perhaps associated mineral structure changes) has a major effect on annealing and diffusion in apatite, and this effect is multi-element, rather than just Cl/F as sometimes assumed for fission track annealing. It is a good idea to promote an analytical protocol of measuring a wide range of elements, rather than just Cl, or a proxy such as Dpar, as these data may be useful in future for annealing model recalibration and/or provenance (e.g. O'Sullivan et al. Earth Science Reviews 201, 2020). However, the available model calibrations are not based on a lot of data, and as stated in Carlson et al. (1999) "*in the absence of any physical understanding of why compositional variations impede or enhance annealing, we have little confidence that it can be used meaningfully to predict the annealing behavior of apatites not included in the experiments*". The concern is that these preliminary calibrations are assumed to give us the definitive model, free of uncertainty, and the rather strong conclusions about the inference of thermal histories are based on that assumption.

We agree that we need to focus this paper better to avoid distractions concerning model calibration, which is discussed in our next paper involving natural AFT samples. In this paper, the ability to convert real elemental data into kinetic parameters is not relevant. We have already predetermined that the synthetic sample has three multikinetic populations with different relative annealing behaviour based on what we have observed in natural samples. The question is, "What are the consequences for modelling the thermal history if you have populations with different annealing kinetics?" In principle, can more thermal history information be retained in a multikinetic sample than a monokinetic sample? It must be kept in mind that the quote from Carlson et al. (1999) is an expectation that the empirical model would not be reliable because of limited calibration. This is an inference that was not tested with follow up studies. Unfortunately, this statement may have deterred people from attempting to use the method. It is also worth noting that in the absence of a perfect understanding of *why* composition alters FT annealing behavior does not negate the fact that empirical evidence indicates it is a real phenomenon and does not ultimately prevent it from being useful.

For example, the results presented are based on more or less ideal data with well separated (kinetic/composition and age) populations (as described in 236 to 249). In this case, we can pretty much recover what we started with including heating out needing to specify near surface constraints on the thermal history. Furthermore, the modelling approach implemented in QTQt tends to prefer simple models, conditional on fitting the observed data adequately relative to more complex models. I think this should mean that individual models making up the credible interval range figure 3a will tend to look like the ML

model shown in that figure, and we do not fully recover the thermal history of 20°C for about 500 m.y. prior to the first heating event ) - I would guess it is just the timing of reheating that changes, and probably the same for the second heating event.

Yes, using an ideal data set, we can recover much of the input thermal history except for the low temperature part where the model lacks sensitivity. Although this may appear to be a self-evident conclusion – modelling of multikinetic data is not a straightforward exercise. The ability to recover good solutions depends on the choice of modelling method and on how it is applied. Here, we demonstrate that good quality multikinetic data may preserve a record of multiple heating events under the right conditions. We do not think that this is widely appreciated because people are used to dealing with single populations that are sensitive to a narrower range of the thermal history. This in part explains the fixation by many workers with whether or not AFT samples pass or fail the $X^2$ test — where passing does not absolutely ensure the absence of multiple sub-populations, nor does $X^2$ failure imply a 'poor quality' dataset or offer a direct means for understanding failure of the statistical test, which could occur due to many reasons such as high precision single-grain ages, large analytical sample size, or compositional heterogeneity (i.e., differential annealing). If none of these variables are assessed, dealt with, or ruled out – or the necessary data collected to do so – then how can the AFT community make any real progress towards advancing data interpretation and modeling practices?

The authors then demonstrate that combining these sub-populations and assuming an average composition (but generally fixed) leads to lack or resolution and/or spurious results for the inferred thermal history. The authors often imply that the latter is common practice, but do not really give any concrete examples. I think many, if not most, people working with fission track data are aware of the potential for over dispersed age data and hopefully would deal appropriately with an over dispersed population (using subpopulations based on composition or age, or perhaps remove egregious outliers - also this issue seems to have different significance depending on whether the data are collected with the traditional EDM method or LA-ICPMS, the latter method tends to have greater dispersion but similar central ages.

We think that modelling of mixed AFT populations is more common than realized but that it is unintended. We could cite specific examples but that could be viewed poorly, as we do not think the practice of 'multikinetic misidentification' is carried out on purpose, but instead with this paper and others in the future, we would hopefully draw attention to deeper investigation of complicated datasets by the AFT community. Awareness of overdispersed data and taking appropriate actions to understand it are two different things, with the latter likely yielding to the former in the majority of cases. If this were not so, there would be more examples to point to in the literature. We believe that multikinetic populations are best-resolved using elemental data (which are rarely collected) whereas population overlap is normal when using low-resolution kinetic parameters such as Dpar. If you cannot discriminate between populations, then modelling mixed populations is unavoidable. These issues cannot be addressed here with synthetic data but are the subject of the next paper. We will choose our words carefully here. We want to make the cautionary point that modelling multikinetic data as a single population can distort the results and it is something to take into consideration. In our experience, multikinetic populations are evident in both EDM and LA-ICP-MS AFT data. The reported higher dispersion with the LA-ICP-MS method is not a problem and both analytical approaches and derived data have been shown to provide similar results. The key is to have enough age and length data to characterize the different populations properly. This topic will be discussed in a subsequent paper that uses natural AFT samples.

The idea of AFT ages increasing then decreasing with Cl content was mentioned some time ago, I think by Barry Kohn, who had some data from Canada implying that the age decreases at high Cl (6% ??) which was associated with a change in crystallographic system from hexagonal to monoclinic. Not sure he ever published that though.

We are not sure what the reviewer wants here. We do not include real elemental data so this extra detail would not add significant information to our paper.

Not clear if the sensitivity of the data to composition as proposed is enhanced due to the long timescales, or this is a general feature.

This is a general feature of the data independent of timescale. We chose a deep time problem because it is harder to deal with than a Phanerozoic situation that may have more geological constraints available, and in general, deep-time problems are afflicted by greater uncertainty, where more data would be even more beneficial.

Also you switch between different combinations of AFT and AHe data, but the latter are fairly minor to the main proposition (and their sensitivity is based on the radiation damage model based on a FT annealing model). You could remove the AHe aspect totally and not change the message...expect perhaps in section 5.4, which comes pretty late and out of the blue.

We will try to rework the AHe part. It is true that the most important part of the paper concerns multikinetic FT data. However, we know that it can be difficult to reconcile AFT and AHe data sets for different reasons, especially for cratonic histories. We wanted to present one case where AFT composition may influence AHe modelling. Compositional data are not routinely used for AHe, but the commonly used radiation damage models are based on AFT annealing. If AFT annealing is truly affected by the composition then the radiation damage model should reflect the change in AFT annealing related to composition. We wanted to investigate this end member situation where all of the AHe age variation is controlled by radiation damage just to make a point that apatite composition may affect how alpha radiation damage accumulates. We will add a section upfront in the introduction to clearly outline why the AHe data are included and the reasoning behind the modeling. This should alleviate concerns that those data are a distraction, yet offer seeds for discussion for future work regarding if and how apatite composition affects He diffusion.

Lines 197 (and 213) There seems to be the assumption that AHe ages depend on composition, but is this just because the assumed model for radiation damage is one incorporating a fission track annealing model

The reviewer's statement is outside the scope of this paper but — yes, some published work has suggested composition may influence AHe ages directly but some work has suggested the opposite. No studies show a strong direct connection, but logic and the *ab initio* modeling suggest it should be so. The reviewer's point is a good one in that an additional issue could be secondhand via the kinetic model calibration. But annealing must involve diffusion, right? So the diffusion that heals tracks depends on small substitutions? The jury is still out. In our example, composition indirectly influences AHe ages because the radiation damage model is tied to AFT annealing, which can vary strongly as a function of composition.

Line 214 - QTQt will generate thermal histories regardless of ..data.... This is true of any sampling based modelling approach. The important point is how the generated thermal histories are accepted or not - QTQt effectively uses the ratio of data fit (likelihood) between a current and proposed model, while HeFTy, another piece of software for modelling thermal histories from thermochronological data, uses an absolute approach (p-value as a measure of data fit) for each thermal history. Perhaps accept is a better word than generate (also on 218), but keep the statement about the user needing to assess the output, particularly how well any particular model predicts the observed data.

Okay we will change generate to accept.

Line 249 - 3% seems small for AHe age data ??

We are assuming good quality, homogeneous AHe grains of suitable grain size and 3% is a normal analytical uncertainty without factoring in the Ft correction. We are trying to reproduce an observed date produced from a forward model, we are not modeling real data where data uncertainty may be cause for concern, when we in fact are trying to recover an unknown history. Ultimately the uncertainty should not be a major concern regardless of the value applied — especially in QTQt where better data quality is rewarded.

Line 256 - what happens if you do not constrain the heating/cooling rates ?

It is appropriate to determine suitable boundary conditions that are commensurate with the scale of the problem. The 5°C/Myr rate limit is more than ample for a timescale of 2 Ga. Given the unimodal and relatively broad length distributions and the old AFT ages, it is not possible to resolve extreme heating rates over this timescale. The main effect of not imposing a rate limit is to have longer model run times with the possibility of introducing rapid heating/cooling artifacts. This is especially deleterious for nondirected Monte Carlo schemes such as HeFTy and AFTINV. We will show results for the AFTINV model and discuss required boundary conditions that are needed to focus the model in promising areas of solution space. We chose QTQt for modelling this deep time problem because it has a learning algorithm that refines solution space and model boundary conditions as it evolves. For nondirected Monte Carlo schemes, the model may generate millions of trial solutions and only converge on a small number of acceptable solutions if boundary conditions are too broad. Even with perfect data, it may be hard for nonlearning models to find answers over such large timescales.

Line 257 - "t-T points were only *accepted* if they provided.."..the points are added, but the important step is whether they were accepted or not.

We will change "added" to "accepted."

Line 262 - put the constraint definitions in section 3.1, and perhaps explain what they represent geologically

We can elaborate on the geological meaning of the constraint boxes. However, discussion of constraints does not belong in section 3.1 that deals with forward modelling (no constraint boxes are used here).

Line 264 - the ML model is potentially more complex than the MP model, but not always, and similarly the MP is not always simpler than the ML model...they can be the same.

We will reword to reflect this. The ML model is commonly more complex. The MP model is usually simpler.

Line 285 - not sure what you mean by a simple temperature weighting....the expected model is defined as

$$E(x) = \int xp(x)dx$$

and in this case the p(x) is the posterior. Given that the distribution of accepted models is the posterior then we just take the arithmetic mean of all accepted models to do that integral. The lower temperatures are because the distributions on temperature around the time of maximum temperature are often skewed (to lower temperatures) and so that leads to lower values for the expected (mean) maximum temperature. However, we often see that the duration of time close to the maximum temperature is greater for the expected model than say the ML model. This may tend to compensate a little in terms of fitting the data, but often not enough...you need to look at the predicted values relative to the observations. Note that we do not generally expect sharp V shaped thermal histories anyway (due to diffusion), but that is another issue.

This was a minor point and could be removed outright, but due to poor wording this should have more clearly stated that the averaging effect effectively pulls temperature downward. We will reword. The main point is the bias toward cooler temperatures.

Line 314 - QTQt does not use the central age directly (or even indirectly) as a model constraint. The predicted age for a given kinetic parameter and thermal history is used to infer the equivalent predicted ρs/ρi ratio which is then used in the likelihood function with the measured Ns and Ni values for each grain (see Gallagher 1995, EPSL).

We will reword this statement. The input is the central age. How the model uses that information is another issue. We are trying to avoid going into too many fine details of the QTQt modeling because this information has been published elsewhere.

Section 4.2 -line 300 perhaps just discuss using the correct kinetics (here you choose rmr0, but this could be another parameter, even if less sensitive ?)

The point is that the kinetics are known in advance for this synthetic data set. The Ketcham et al. (1999, 2007) annealing model uses $r_{mr0}$ values. If users specify Cl content or Dpar, they are converted to $r_{mr0}$ values using empirical correlations for the purpose of the kinetic model. We chose $r_{mr0}$ based on our experience that elemental-derived $r_{mr0}$ values provide better kinetic population resolution than Dpar or Cl content alone. We cannot show that in this paper because we do not include real data for a natural sample.

Line 317 - when using the average eCl 0.213±0.373 in QTQt, this implies that you let the kinetic parameter vary as part of the inevrsion...what was the distribution of the accepted values ? If skewed to a higher or lower value, that may be informative concerning the sensitivity of the different subgroups of data to an average/common kinetic parameter.

We can show this information in the supplement if necessary. Yes, the kinetic parameter was allowed to vary during the inversion, however regardless of skew in either direction, the point is that you could assign any kinetic value here within reason, *but more importantly*, if a multikinetic sample is misinterpreted as a single population and the true history is complex (like we see in our example), then the fit to the observed data may be perfect, but one would never recover a history that closely resembles the true history. Therefore, as the reviewer mentioned previously the model assumption would be incorrect, but in this case both the t–T model AND the assumed single population are both incorrect. There seems to be a misperception in the reviewer's statement because the sensitivity of the different subgroups are lost entirely when combined into a single, monokinetic population — hence the point of this exercise. Combining the data yields some 'average' AFT age and some average kinetic value that removes t–T sensitivity. It is worth noting that *not all AFT samples are multikinetic* and we are not claiming this. Each natural sample is unique, and some may be multikinetic and others may not be, but if compositional data are not collected in the first place, how can this be known or addressed?

Line 318 - perhaps some examples of overdispersed data treated as a single population. The impact of this is likely to depend on how overdispersed and why...failing the chi-sq pvale = 5% test is not necessarily the definitive indication (e.g. we can pass at a level of 5.0001, but fail at 4.9999). Leaving aside analytical problems, dispersion that is a real if sometimes unwelcome signal could be due to compositional effects and discrete provenance related age populations (for which compositional ranges may be similar).

We will reduce some of this section. Based on the reviews, we think part of this discussion is best left to the next paper that includes real data for natural samples. Without including real data, we cannot show why we think unresolved, mixed AFT populations may be a larger issue than recognized. However, we do provide an example of overdispersed data being treated as a single population in our model examples. See our comment above regarding the $X^2$ test.

Line 320 - we do not necessarily have to formally identify discrete groups with mixure modelling, but just divide compositional range into subgroups and use the appropriate values (e.g. as Geotrack seem to do for Cl binned at 0.1wt % intervals)

Yes, but how many people do this other than Geotrack? That isn't clear in the literature overall and it isn't very clear at all how Geotrack carries out thermal history analysis. We think discrete models work better than a more continuous model for the natural multikinetic samples we have observed. However, we cannot demonstrate this here with a synthetic example, so it is beyond the scope of this paper. Our synthetic example is based on features we have seen in natural samples. Dealing with the distribution and allocation of track lengths in either a discrete or continuous approach is the most important factor to consider, but is outside the scope here.

Line 330 - it is not that QTQt failed to reproduce the true AHe dates...it is because the wrong choice of model prevented QTQt from doing so....

We will reword to explain that the true AHe dates were unable to be reproduced due to incorrect data treatment, i.e., incorrect model choice (see 'line 317' comment above as well)

Line 359 - I think that Geotrack do use compositionally discrete modelling for their routine AFT studies, but we rarely get to see the predictions for their preferred models.

That may well be but unless it is clearly documented in the literature then we are not privy to the details and can only speculate on how things were done. We still think it is fair to say this is underutilized in the scientific literature.

Line 388 - data quality is important too...

Sure, we did not list everything. Data quality is a factor but it is a function of many things. You need enough measurements to define populations, you need representative measurements for each grain and this could be affected by zoning, stressed grains with dislocations, problems with analytical procedures, etc.

Line 391 - what do you mean by more extensive ?

Greater time-temperature ranges. We can state this more clearly.

Line 396 - as I said above, perhaps we should model each grain with a specific set of kinetics ? This does no necessarily require running N annealing models for N grains, but perhaps 4-5 and we can interpolate the results (e.g. predicted ages and length distributions) for intermediate compositions.

We think the discrete model is a better way to go. You have well defined discrete populations. Why pick something in the middle and subdivide these well-defined populations? What is to be gained by further subdivision? If coherent age and length populations appear on plots of AFT parameters versus kinetic parameter, why resolve populations to a finer scale than necessary? In any case, this is a synthetic sample with predetermined properties. We also stress that arbitrary division into populations may be problematic depending on how this is carried out. With real samples we use radial plot mixture modeling to first identify discrete populations and in nearly all cases these populations coincidentally or naturally align with breaks or divisions within the AFT age data with respect to apatite composition. This agreement suggests to us that there is real kinetic/differential annealing giving rise to variability in single-grain ages and that those ages in turn correlate with changes in composition. After all, if single-grain dates correlate with composition and the t–T path of the sample caused differential annealing, we should expect a $X^2$ test failure and mixture models should reveal discrete subgroups.

Line 401 - without necessarily requiring

We can add this qualification.

Line 407 - what was the first hand ? I had forgotten by the time I got here.

We will tighten up the text so you remember.

Line 413-415...not sure I understand this...explicit condition on the model prior ? Perhaps you mean on the model sampling.

Yes, will reword.

Line 415 - not correct...it is the posterior that will be lower I think, rather than the likelihood. At least the acceptance of models is based on the posterior

We will rephrase this to be correct in our Bayesian language

Line 418 assist - use focus perhaps.

We will change this.

Line 421 - proper data interpretation is vague...you mean some of sub-population classification based on composition, age dispersion ?

We will replace proper data interpretation with kinetic population interpretation.

The discussion of precision v accuracy based on the credible intervals does not really take account of the form of the individual thermal histories...although they may define an envelope that seems consistent with the known true answer, the actual thermal histories may not really capture the true thermal history as well.

Yes, we know that QTQt calculations differ from models such as HeFTy and AFTINV where the plotted individual solutions are required to fit the data at a specified level of significance. Depending on how plots look we may add all individual paths to show how and where the Expected ± 95% envelope falls.

Line 429 - again, it is difficult to assess if we really resolve the true thermal history from the credible intervals..they do not tell us about the variation of individual thermal histories...so the earlier reheating event may start at different times, but we do not necessarily resolve the period sitting at low temperatures for a long time.

Yes, we know the issue with the individual solutions. The important point is the main features of the forward model thermal history are recovered by the Maximum Likelihood and Expected models. See point above.

Line 434 - more complex thermal histories were accepted or retained rather than added.

We will change added to accepted.

The rest of this paragraph to line 439...when you allow more complex models that do not improve the data fit to be accepted, you start to sample the prior more extensively, i.e. you tend to fill up the prior box in those parts of the time-temperature space where the data do not care what happens. For example, prior to a reheating event, the temperature can be pretty much anywhere from the reheating maximum value to the lower limit of the temperature prior. This does not mean we resolve the thermal history there. Almost the opposite. All we resolve is that the temperature has to be lower than the subsequent maximum.

Makes sense. The model lacks sensitivity at low temperature and it is allowed to fill up the space. We will modify the text.

Line 443 - Apfu Cl ?..if you add normally distributed noise, or even uniformly distributed, with the same mean value as the true model, it is not a great surprise that the expected model and credible intervals do not change much as we will tend to fit the average values of the data (which will reflect average values of the kinetics, which will be sampled on average to give the correct value...if that makes sense).

We will add eCl after apfu in the brackets. Kinetic parameters are relative. If the mean is similar to the "true" kinetic value used in the forward model, the inverse model will converge to be close to the input values. If mean values are not close to the input value, the model may converge on a different value but it will need to offset the remaining kinetic parameter to satisfy the observations. The result is that model temperatures can shift upward or downward but still converge to the same relative temperature history. Depending on how input ranges are specified, final kinetic parameters can be anywhere within the input range. The model will have trouble converging on good solutions if kinetic parameter ranges are too narrow to accommodate the required adjustment to kinetic parameters. None of this changes the conclusion of this paper that multikinetic data may contain a record of multiple heating events.

Line 448 - this acknowledgement of imperfect kinetic models should be stated much earlier, and ideally quantified somehow..(i.e. the recalibration exercise I mentioned above).

Recalibration is not important for this paper because we are using synthetic data. We assume the kinetic parameters are known and we investigate whether we can recover important thermal history information from a multikinetic sample using inverse modelling. This drives home the point that our current kinetic

models do have some validity. All AFT modelling suffers from the same uncertainty on kinetic parameters. The most important aspect of a multikinetic model is the relative behaviour of the system. Calibration affects the absolute model temperatures. Relative kinetic behaviour preserves important details of the thermal history and this is important distinction will be discussed in the next paper.

Line 456-58 - the data quality (and the associated information on the thermal history) is first order in terms of what we can resolve. If the data are poor, scarce, we want more uncertainty.

Yes, data quality is critical. Uncertainty increases as data quality degrades. There are many factors influencing model results. These should be explored in future papers after as we learn more about natural multikinetic samples.

Section 5.3 - this deals with a separate issue to the main thrust of the paper, and adds to the feeling of stream of consciousness. I think it is not just with QTQt that we need to assess the effect of constraints on the final results and this has been said before in the exchanges in Earth Science Reviews over the last few years, and still ongoing. Any model result is conditional on the assumptions made to obtain the result. Thus a no constraint model can often be relatively boring (i.e. linearish cooling) but if the data can be adequately explained that way, then it is useful to know and can be considered as an end member model. Adding constraints is not a problem, but these need to be justified and this needs to be clearly stated in any study, preferably with some assessment of the confidence in a constraint. Additionally, when a thermal history is composed of linear segments joining up constraint boxes and we fit the data, this does not mean the constraints are justified...just that the data do not contradict these constraints and they do not require more complexity than imposed by the constraints.

A combination of inverse models and targeted forward models, often based on the inversion results, can be useful to deal with many of the problems discussed in this section and you do say that ..but you could just say that more concisely.

Yes, we will streamline this. There are many different situations and it is hard to generalize based on modelling one synthetic data set. There are different modelling strategies. Nondirected Monte Carlo models such as HeFTy and AFTINV usually require constraints in order to restrict model search space or they may have great difficulty converging. We just wanted to point out that constraint boxes might not be helpful under certain conditions. If multikinetic behaviour is unrecognized due to a lack of compositional data, for example, then thermal histories may be distorted to fit the data (see above). Use of constraint boxes will not lead to a better solution under these conditions.

Lines 474-476...not clear what this means...especially linearizing bias and Bayesian treatment of user constraints ..?

We will remove this text as it is not entirely necessary and as we can see from the comment that it can be confusing or misleading. What we were attempting to point out is that if 'more complex' models are prevented from being accepted (regardless of likelihood) then if there are two boxes placed within QTQt, the tendency will be to connect the two boxes with a simple 'linear' history segment, **IF** the data do not require a more complex path to provide a better fit. This does not mean that linear segment is legitimate or 'real' — which is where the point regarding a basement nonconformity being subaerial 100s of Myr before the geologic evidence (and a small constraint box) would suggest. This is likely an artifact of thermal history construction. Without boxes, the model fills the low sensitivity, low temperature space. Surface temperature can occur anywhere in the interval. When all solutions are forced to be at surface early on then temperatures ramp up because of the way temperature histories are constructed. The input history is very difficult to fit at low temperature because it is hard to have an effective heating rate of zero persisting over such a long time interval unless you force the model to do so. You can fit the data without this requirement.

Line 494..I agree with the sentiment...heralded perhaps better stated as imposed under the guise of geological evidence. This goes back to what I said above - there is not a problem using constraints/forcing

a model, but the results are conditional on these and there are generally other models that can fit the data without, or with different constraints. Again, it is how valid the constraints are that is the big question.

Yes, we agree constraints have their place. We will rework the section to make things clearer.

Line 504...again Geotrack seem to use 0.1 Cl wt% bins when modelling, but we never see their predictions. You may add here that limitations arise when modelling samples independently, especially in boreholes. It is clear that jointly modelling multiple samples (plus the mutli-kinetic approach for each sample) is better - data noise will tend to cancel out (if it is random) while real signal (the thermal history) should be reinforced. See Gallagher et al EPSL 2005 for an example with synthetic data too...

We expect models that use 0.1 wt% Cl bins will give different results than those that model discrete multikinetic populations. In the former case, kinetic parameters are expected to vary smoothly. We do not see this in most of our multikinetic samples. Instead, there are significant differences in kinetic behaviour between discrete populations. We also do not think a single parameter like wt% Cl will properly account multikinetic behaviour. This topic cannot be discussed in the current paper. It needs to be mentioned in the context of naturally occurring multikinetic populations that is the subject of a future paper.

Liner 509-510...what if the true thermal history is simple cooling ?

If that is the situation then inverse multikinetic modelling should yield linear cooling histories. We have unpublished natural examples where simple cooling can explain observed multikinetic data. In other cases, complicated histories are required (consistent with the geological complexity of the regions from which the samples were obtained).

Line 513- not sure what you mean by universal slow cooling suppositions...there is nothing forcing cooling in QTQt, apart from perhaps have a sample at surface temperature today and letting it start hotter...if the data are happy with that, then why not ? If the data need more complexity the thermal history should adapt.

This may be true for QTQt. However, other models such as HeFTy and AFTINV can be used to enforce continuous cooling, or enforce anything for that matter. These models may fit at the threshold of acceptability (and therefore cannot be rejected as a possibility) but may not be able to fit the data closely. The question is, "should we prefer the simple model that barely passes and has trouble finding solutions or a somewhat more complicated model that fits the data closely and converges quickly?" Complexity often yields better fits to observed data. As sample quality increases, it usually becomes easier to answer this question. With enough data, the linear cooling model may not reach the acceptance threshold. In any case, treating multikinetic data as a single population may lead to a significantly different and more simplified thermal history. The comment about *"universal slow cooling…then why not?"* highlights the issue cropping up in the literature with the use of QTQt as a black box where people dump in lots of data and hit "go" and especially over long timescales, these data can often be adequately reproduced under linear cooling assumptions. This doesn't mean linear cooling is valid. This is a difficult problem to address.

Much of section 5.4 is relatively speculative but more importantly deviates from the main message of the paper concerning FT annealing and left me a little confused. The last paragraph is OK...but I think you could drop much of this and perhaps save it for another paper....but that is up to the authors. While I agree that allow an effective parameter to vary in the modelling, as QTQt allows, can be useful - the paper by Ricanati et al. demonstrated that yes we can improve the fit to the data by having a wiggle factor for each grain diffusivity. As Samuel Karlin said, 'The purpose of models is not to fit the data but to sharpen the question'...so demonstrating we can fit the data with an additional factor is not really the solution, but suggests we should look for a physical control on that factor. Given the arguments here, I would say go and measure apatite chemistry to demonstrate that there is some correlation of age and chemistry or effective diffusivity then you are on to something (perhaps this will be possible with the method, Pickering et al. mentioned). For me, as it is, the sceptic will take this section as special pleading for a control on AHe date dispersion that is neither understood nor constrained. Those adopting the averaging strategy for AHe data

will just say we do not understand enough to do anything more sophisticated and carry on. I would add a caveat to the Karlin quote too....not fitting the data means we need to ask different questions.

We agree that we need to scale this back and save it for the next paper where we can have more in depth discussions around multikinetic interpretation and modelling. We also agree that we are only showing one possible reason why AHe data may be difficult to model and that much more work is needed to resolve the problem. We will focus the paper more on the key points we want to convey. We would also add that if the thermochronology community does not adequately understand the problems surrounding AHe data and workers have to manipulate or massage dates through averaging, 'culling outliers', or 'eU binning' should papers be published utilizing these data or should more work be done to address these issues head on? This aligns with the reviewer's sentiments or caveat to the Karlin quote.

Line 582...Steve Bergmann was often insistent on the importance of OH as a control on fission track data....not sure if he ever published anything though.

We think this is well established in the literature and that OH is an important factor and it should be included when obtaining detailed elemental data for constraining kinetic parameters. That is why electron microprobe analysis is preferred over LAICPMS elemental data. Carlson et al. (1999) and Ketcham et al. (1999) noted that it was important enough to be a separate term in their empirical $r_{mr0}$ equation. They also noted that $r_{mr0}$ correlated better with OH than Cl or Dpar. This topic is peripheral and better discussed elsewhere.

Line 594 - what recent publications ?

The section on AHe will either be removed entirely or greatly reduced and reworked so these points may no longer be relevant. Also the AHe data will be downplayed in terms of significance and discussion. The relevant citations being mentioned in the previous section being Gautheron et al. 2013 or Gerin et al. 2017, Recanati et al 2017 and Powell et al. 2017 and 2020 for looking at both AFT and AHe composition/kinetic variability together.

Line 595 - not misfit, but poor fits.

We will change this.

Line 596 - degrades in what sense ?

This is a sample specific result. It is hard to generalize. For our case, it results in a simpler thermal history or a change in the timing and magnitude of thermal events relative to the input history (if constraints are enforced).

It may be the paper may be better concluded by adding a series of recommendations on analytical practice/protocol and then modelling strategies.

We cannot say too much here because we are not providing a detailed description of how to interpret multikinetic data for natural samples. Much of this will be dealt with in a future paper.

Fig 1 - perhaps put the 2 constraint boxes on the forward thermal history. They are a little strange as constraints as we might expect the constraint to be the stratigraphic age of sediment deposited at the time of the start of the heating events, rather than some time prior to the heating event.

The constraint boxes do not belong on this figure. They were not used to generate the forward model. They are considered as something inferred from fragmentary geological evidence. We can elaborate on what they represent geologically.

Fig 3 - the credible interval ranges are fine as presented in this figure, but it may be useful to put the sampling of individual models and/or the marginal distribution (the coloured plot from QTQt for a given sample thermal history) in the supplementary - then we can see how many models actually start reheating at 1200 Ma for example.

We will add more QTQt output to the supplement to assess overall model behavior.

**Reviewer 2: Anonymous**

The study is suitable for publication with minor revisions but it could be substantially shorter.

We agree. Some of this material is better suited for our next paper that uses natural samples to illustrate the multikinetic method.

However, a resolution test is only really valid for the specific test at hand. It is clear that by using all the data in the correct way, the correct path can be recovered for this specific path, but this does not mean that multikinetic data is important for all time-temperature paths. This is an obvious problem with this type of study and unless the resolution tests are for a specific problem at hand, it is unclear what we really learn from these tests.

The point regarding multikinetic data being important or not for all t–T paths is mostly true (see comment above re: line 317 from previous reviewer) but using "all data in the correct way" is easy to state but nontrivial in practice. What researchers consider as 'all of the data' and 'correct' are entirely subjective. Synthetic resolution tests are used in a variety of Earth Science disciplines to demonstrate sensitivity (or not), so we are not sure why the reviewer thinks nothing is learned from these types of tests. Using similar logic, why do thermochronologists carry out forward thermal-history models when a nearly infinite range of possibilities exist in most instances? The obvious answer to is to determine endmember cases and whether and how data are sensitive to imposed t–T conditions, which is in essence what we are doing. Numerous factors determine how effective multikinetic data will be for resolving thermal histories. This paper only shows results for one specific example. The purpose of this paper is to show that under certain circumstances it is possible to recover information on multiple heating events from a single multikinetic AFT sample. We want to demonstrate this for a synthetic sample with a known thermal history. Therefore, when real data are modelled we can demonstrate that the concept has been shown to work. At a past conference, we heard people express doubts that multiple heating events could be recovered from a single multikinetic sample. We want to dispel this notion by running synthetic tests to show otherwise. We will revise our introduction to more clearly state this. Inferences of system behaviour based single monokinetic AFT populations are not translated easily to the more complicated multikinetic system. We also want to show that thermal history solutions can be altered significantly by not recognizing multikinetic behaviour. Again, the degree to which this happens will be sample dependent. It is impossible to show all cases, so we just want to demonstrate a proof-of-concept and get people thinking about multikinetic behaviour.

For example, if the burial conditions were ever so slightly different, the tests may imply that AHe data are more important that AFT data or something like that. This point should be acknowledged and maybe the details of why the forward model path is as it is should be discussed.

We can add more text to explain why we chose our example, but we also believe that the thermochronology readership would intuitively understand the point the reviewer is making. There are always exceptions to every case, but this does not negate overall trends to be deduced from data.

It is also unclear whether we are learning something about the data or something about the specific algorithm used to interpret the data. One of the attractive things about QTQt is that you do not need to specify the number of nodes in the inversions or limit the rate of the cooling. However, the authors stress the importance of the maximum likelihood models and not the expected or maximum posterior models that benefit from the reversible jump component of the algorithm. No Bayesian statistics are required to find a maximum likelihood model, so it is unclear why QTQt is used.

We firmly believe we are learning something about the data. We chose QTQt for this deep-time problem because it is a learning algorithm and therefore it is more efficient at finding solutions when constraints are few and time intervals are large. Nondirected Monte Carlo schemes like HeFTy (and AFTINV) need to have appropriate constraints in place in order to limit the model search space. Without the right constraints, these

models will have difficulty finding solutions and they may only converge on solutions that are at the threshold of acceptability. Typical applications of HeFTy are run for a specified number of forward models. Even with ideal data, this may result in a few or no acceptable solutions for the current problem unless suitable constraints are imposed. AFTINV is typically run to obtain a specified number of acceptable solutions (usually 300). In this case, even with constraints, millions of forward models are needed to converge on a set of solutions. The multikinetic data contain information on the thermal history but the ability to extract that information is model-dependent. We will include results obtained from AFTINV to illustrate this point.

For example, in Figure 3e, the ML model is actually outside of the credible intervals around the EX model. This is probably an advantage of the EX model because often the ML model is wildly complex. If the EX models are taken as a suitable compromise between averaging data and resolution, all of the first 6 models in figure 3 (3a-3f) look very similar. This highlights some of the points made by Vermeesch and Tian (2014) and some discussion about which model to concentrate on might be useful.

We emphasize the result that provides the closest fit to the data. It must be remembered that individual solutions that define the credible interval may not fit the data very well and we already know what the history should be. The peak temperatures of the ML path often fall outside or on the limits defined by the credible intervals in reheating scenarios, and of course the credible limits are 95%, not 100%. However, it is the point about the data fit that is sensitive to that peak temperature that is important. The reviewer has a misconception regarding the implementation of QTQt in these modeling scenarios. We only allowed 'more complex' models to be accepted if they provided a better fit to the observed data, whereas classically this was not an option in previous versions of QTQt, and models were accepted regardless of likelihood, which meant that algorithm using the reversible jump aspect of the MCMC to ultimately 'figure itself out' with respect to which models were better fits to data. This approach results in a more thorough exploration of t–T space and more easily shows where data are or are not sensitive, but results in wildly complex ML paths. In this case the MP path is often preferred due to (in general) nearly similar log likelihood but with simpler (i.e., more realistic) and fewer t–T points. A general issue with the EX model is that in cases of reheating, the overall form is okay but it can yield poor fits to the data due to smoothing.

An additional set of models that use predicted data using one kinetic population and then invert them with the correct single population kinetics might help demonstrate the value added in having samples with multikinetics.

We have run the individual populations and we do not think it is necessary to show all results, but we will add a figure or additional fig. panel that shows the QTQt results for each kinetic population individually to show where the data are sensitive. From AFTINV modelling (we will add this), we can get an estimate of model annealing temperatures and it becomes evident that population 2 is critical for defining the first burial peak. Estimated total annealing temperatures for population 1, 2 and 3 are approximately 80°C, 110°C and >250°C, respectively. Therefore, population 3 offers little constraint on the two burial heating episodes. Population 2 is most sensitive to the first heating event and has some sensitivity to the second event. Population 1 has been thermally reset during the first heating event and it retains a record of subsequent cooling and reheating by the second event. We will discuss this along with the AFTINV model results.

I also think that in many cases detrital samples have multikinetic data because the apatites are from different source areas. In turn, there is the basic assumption that it is appropriate to treat all these crystals as having the same thermal history. In fact, the crystals may have distinct thermal histories that may or may not be important in the interpretation. For example, Carter and Gallagher (Carter, A. and Gallagher, K., 2004. Characterizing the significance of provenance on the inference of thermal history models from apatite fission-track data-a synthetic data study. SPECIAL PAPERS-GEOLOGICAL SOCIETY OF AMERICA, pp.7-24.) describe this issue for the case of AFT data and Fox et al., 2019 (Fox, M., Dai, J.G. and Carter, A., 2019. Badly behaved detrital (UâǍˇ RTh)/He ages: Problems with He diffusion models or geological models?. Geochemistry, Geophysics, Geosystems, 20(5), pp.2418-2432.) describe this for AHe data. In many cases, it is unclear if sufficient temperatures have been reached to effectively remove all the previously accumulated "age" and any additional factors that may control age accumulation.

We briefly discussed this point in the text regarding multikinetic data and detrital sources. This is definitely an important point when dealing with natural samples but it is not relevant for our synthetic example that was created using a common thermal history. We can certainly emphasize this point when describing our example. Based on our large set of unpublished Phanerozoic detrital samples from northern Canada, we can say that this issue is recognized easily and it is not common, at least among the samples we have examined. Most of the data can be modelled using a common history. It is something to consider but it is not a large enough problem to discourage people from taking a multikinetic approach. This issue for AHe data in the cited example likely exists but would be hard to decipher in practice, given all the other baggage AHe carry. In addition, it should be noted that within QTQt the problem of 'inherited' pre-depositional history can at least be partially dealt with by adding a single t–T point prior to the depositional age, which can be independent for each kinetic population. This in turn means that the thermal histories are allowed to diverge prior to deposition and can vary, whereas they converge post-deposition.

It is not clear how noise is incorporated into the analysis. On line 236, are the dates of each individual measurement, for the case of AHe, shifted by a specified amount or are the uncertainties on the true age set based on the noise value. I think this is clarified on 249 where the ages are the correct age with an additional uncertainty. It would be interesting to know what happens when the input ages are drawn from a distribution given by the true age and a 10% uncertainty. But the 3% errors seem a bit small. Similarly, if a larger dataset of say 10 ages were measured, a larger spread in eU might be predicted and this would have important implications for the amount of information added by the AHe ages.

The uncertainties were just increased on the AHe dates. The point of our study is to see whether we can recover thermal history information from a high-quality synthetic example. We can always add more noise and decrease the thermal history resolution. There are many ways to alter model resolution by adding noise or changing model parameters. We don't see the value at this stage in a more rigorous sensitivity analysis until there are more published examples of multikinetic data. The suggestions regarding variable eU are valid but outside the scope here. We wanted to make the simple point that for AHe dates *of the same grain size and eU* that large age variability or dispersion can be a result of *composition* (i.e., $r_{mr0}$, if composition is believed to have an effect on He diffusion within the AFT annealing kinetic proxy framework). Too often in published papers grain size and eU are the sole factors discussed – what happens when there is persistent dispersion and no date correlation with grain size or eU? There are many 'whatabouts' we could discuss that are outside the scope of this paper.

Line Comments 63: mean etch figure width 68: I think you need remind people why AFT ages are ages and AHe are dates and not ages. My understanding is that date is preferred for AHe to reflect the idea that this does not correspond to a specific event. Surely this is equally true of AFT central ages? Why not just use age to be consistent with the AFT literature?

The only reason for AHe 'dates' versus 'AFT ages' is historical only. Ages have implied geologic meaning whereas dates do not. We will use 'apparent age' for He data discussion.

85-88: This sentence is a bit long.

If we can shorten and convey the same message, we will do it.

256: It is not clear how the more complex models were rejected here.

We will clarify this, but it is an option implemented within the newer versions of QTQt. See comments above on this point.

285: "because the EX model undergoes a simple temperature weighting in QTQt,", this is not correct. The model integrates all parts of temperature weighted by posterior probability.

We will modify the description. This was pointed out by reviewer 1 as well.

414: check that this is actually likelihood and not posterior probability.

Ok. We will verify this.

415: greater uncertainty – this should really be greater certainty to mirror the idea that accuracy is closer to the true solution.

Yes, we will change this to maintain consistency in what we are saying.

419: there isn't really an envelope of accepted models in a QTQt model. There will be lots of very bad models accepted during the burn in phase for example, and in order to approximate the extremes of the posterior distribution, bad models need to be accepted.

Yes, we will clarify this point, that was more of a brief slip into the random MC realm. For other models like HeFTy and AFTINV, the envelope contains accepted models but QTQt is different.

442: Please be more specific about how noise is added.

We will elaborate. This point was also discussed by reviewer 1.

500: I guess that the conclusions of Green and Duddy would probably be correct if the temps during the second burial peak event were a bit higher.

Yes, the sequence and magnitude of thermal events is critical to whether or not information is retained. Clearly if the hottest event occurs last, then the prior history is obscured or information lost entirely. We are pointing out that it cannot be a general rule that you are unable to distinguish multiple thermal events using multikinetic data (i.e., their statement in light of using multiple thermochronometers means that they are only correct some of the time, yet they state this as certainty all the time). It depends on the nature of the sample and the thermal history. The large number of unpublished samples in northern Canada that retain a complex record suggest that this situation is common. We hope to demonstrate this in future publications.

542: Has CRH been defined in this manuscript? It is probably worth describing what that is.

Yes, this should be defined.

**Reviewer 3: Richard Ketcham**

This is a tricky kind of study to do, because it's difficult to generalize the problem – in which cases does multi-kinetics matter, and in which can it be neglected without overt penalty? For example, if there is fast cooling, subtle changes in kinetics will not matter too much. Put simply, careful attention to kinetics is likely to be most important in cases of long persistence at, or reheating to, a temperature range that differentiates the thermal responses of the grains present, and thus the ages and lengths recorded. It might be best to state this up front, and then pose the subsequent tests as a demonstration of that principle.

We agree completely with this. We will rework the introduction and more clearly state the rationale for what we are doing.

The synthetic data set is a bit over the top, in terms of quality. There are three kinetic populations, all equally represented in terms of grains and tracks. The synthetic t-T path has been designed to just touch into the lower part of the PAZ once for each of the two lower-resistance populations, in the necessary sequence for evidence of each to be preserved. More importantly, the authors say they add an "appropriate level of noise," but don't specify what that is, or how they did it. The age uncertainties on each population are all less than 5%, and all three populations have chi-squared probabilities of 100%, suggesting that the single-grain ages are under-dispersed compared to a true natural sample. This is borne out in the numbers in their Appendix; all grains but one are within 0.5-sigma of the central age; the one exception is almost exactly 1-sigma. This is massively under-dispersed compared to what one would expect with a natural sample (i.e., a random sample from a normal or Poissonian distribution), suggesting that something about how they generated these synthetic data was a bit off. This needs to be redone.

We purposely chose an ideal data set to demonstrate that important details of the thermal history can be retained in multikinetic samples under favourable conditions that are devoid of factors that will degrade model resolution. The three populations are equally balanced. Otherwise, one must investigate how changing the distribution of data among populations affects results. Adding some more noise to the data will not change the basic conclusions unless the noise is so large that populations cannot be clearly resolved or population ages are highly skewed. Then criticism could be targeted at how well distinct populations can be resolved. The endmember is an important reference point to illustrate that a record of multiple heating events is preserved. As you add more noise, the record will become less well resolved. The problem is, "how much noise do you add from the universe of possibilities?" We did not intend this to be an examination of the ability of models to extract information under various conditions. The degree to which you can extract information from real samples is a topic of our next paper that uses natural samples. This synthetic sample is drawn from a range of possibilities, so it is difficult to generalize how models are influenced by adding age dispersion. Is the dispersion uniform around the population age or should it be skewed? How much? There are too many other factors that affect model resolution (see discussion above in overview comments to reviews) so an investigation of model sensitivity to data dispersion is only relevant for this synthetic example and for the QTQt model we are using. We think these types of model investigations are important, but they should be informed by real examples and more multikinetic data need to be published to guide these investigations.

The result is a data set that grabs the viewer by the lapels and shouts "multi-population, multi-kinetic." Perhaps this was the point, but it does not make for effective advertising, as real data will never be this clear. It also does not make for a realistic test of the ability of thermal history inversion to read the history. They need to run the test with a normal degree of dispersion, and then it might be fun to run one with some excess dispersion, such as by adding some dispersion into the input kinetics, or adding another couple of small populations at different kinetics that are unidentifiable as populations because there are so few grains.

We think this should be done in a more comprehensive paper in the future that is informed by what is observed in real multikinetic populations derived using detailed elemental data. There are not enough published examples to infer how well multikinetic samples can be resolved. A large unpublished data set for northern Canada shows a broad range of situations with many examples of well-resolved kinetic populations that will be published in the future. We agree that there are many factors affecting model resolution that are worthy of investigation. However, the ability of models to extract thermal history information under variable sample conditions is not the point we are trying to make. Thermal history resolution depends not only on data quality but also on the modelling strategy. Both HeFTy and AFTINV are nondirected Monte Carlo methods that share many similarities. We will add AFTINV models results for the synthetic sample to show that, even with near perfect data, modelling is not straightforward and convergence to good solutions is not guaranteed.

We would also turn this critique around and ask if we showed real data, would the value of, or even the ability to carry out multikinetic interpretation convince the uninformed skeptic? Insofar as the age dispersion for the **overall AFT sample** — we are well within the range for 'normal' natural samples. The assessment of age dispersion and $X^2$ probability is done first with the entire AFT sample and then the subpopulations are defined, and their individual dispersion is then assessed (we are just showing this in reverse order). Yes, the individual populations have extremely low dispersion, but in the overall multikinetic scheme this does not alter the modeling or the results.

I think they could also have done more with testing along the lines of what's in Figure 5C, where they deleted a population. First, I disagree with the authors on that result – I think the penalty is surprisingly modest, to the extent of bringing up the question of how important that middle population is. If they had, say, a constraint for the depositional age of the initial sediments of the first burial episode, they could probably do without the second population entirely. This may be foreseeable if the lower part of the PAZ for the most-resistant population overlaps with the upper part of the PAZ of the least-resistant

one, thus providing effectively continuous coverage. It also gets to a practical matter – if kinetics appear messy in the data, with lots of overlap (which there usually is), can the major information be extracted by concentrating on the end-members present? This would be a useful question to answer, or at least address. Second, another version of this exercise might allow the authors to run a sub-test – say, omit the highest-resistance population, see what they are missing, and then run the two lower-resistance populations as a single one, and see how that (also) messes up, in a 2-population case. This would be a simpler case to reinforce the general point that reheating to the PAZ is when multi-kinetics gets important.

Various models were run but we only included the most significant results. We do not share the reviewer's view that population 2 is not important. This is apparent in the difference between the ML and MP paths when that population is removed, but this fact is obscured by the overall model ensemble. We are going to add in a figure/panel that shows the QTQt results of each individual kinetic population under the same prescribed thermal history to demonstrate the sensitivity of each population. Modelling of individual populations confirms that population 2 contains the most information concerning the first heating event. If needed, such information could be put in the supplement. This result is confirmed by the approximate total annealing temperatures for each population that were estimated using AFTINV model results. There is much less overlap in annealing behaviour between population 1 and 3 than the reviewer thinks. The total annealing temperature (when tracks < 2 microns disappear) for population 3 is close to 290°C whereas it is approximately 80°C for population 1. Population 3 has low sensitivity to the two heating events. Population 1 was reset by the first event and records subsequent cooling and reheating during the second event. Population 2 has a typical apatite composition with an annealing temperature of approximately 110°C. It experiences strong annealing by the first heating event and has some sensitivity to the second event. Therefore, populations 1 and 2 retain a record of the two events.

The authors lost me a bit in the discussion of rmr0; this section needs to be shortened and clarified, while keeping the important parts. I have heard informally that people comparing the 2007 and 1999 models in some situations have noted divergent behavior, and have tended to prefer the 1999 one, but I have not seen a quantitative exploration of why this might be the case. The authors claim that the difference may lie in the 2007 version weighting more resistant, and Cl-rich apatites versus apatites that feature cation substitutions. This may be true, but I don't know; it's not clear that augmenting one aspect of compositional space necessarily diminishes the influence of another. Alternative possibilities include other unknown or unexplored incompatibilities between the Carlson and Barbarand data sets, or that the empirical fitting method is oversimplified. In particular, the 2007 model has a relatively compressed total thermal sensitivity range (maximum Tc = 180C in 1999 (note 2006 erratum), 160C in 2007) because of how the Carlson and Barbarand data sets interacted. It's also worth being very clear that the kinetic meaning of rmr0 differs between annealing models – for example, Durango is 0.827 in the 1999 paper and 0.797 in 2007 (rounding down to 0.79 after applying rmr0+$\kappa$=1.04), but these values lead to the same closure temperature in their respective annealing equations.

We agree. We will shorten this. These comments are better explained by reference to real examples where elemental data are used to calculate kinetic parameters. See response to comments from K. Gallagher as well.

The authors could also be a bit more clear when they discuss constraint boxes. To the extent that imposed constraints embody reliable independent geological information, they are *always* proper to add. Similarly, if the geology justifies them, there is no such thing as "excessively tiny" (line 473) – they should be the exact size the geology says they should be. In fact, all paths that go outside those constraints contradict the geology, so why should they even be considered? The two downsides the authors cite seem to some extent like red herrings. First, there is the issue that QTQt seeks the simplest paths, which means it minimizes the number of t-T points. By making a constraint box, they are basically telling one of those few points where it must be, reducing the freedom of the others, and giving the appearance of precision. However, the broader credible interval given by an unconstrained QTQt is not

a better reconstruction of the thermal history, because the parts of the envelope that overlap the otherwise neglected part of the true path are based on paths that violate the local geology. (Naturally, the proper solution is to use HeFTy, which will broaden envelopes where the thermal history is poorly constrained by the data ;-) ). Second, the authors caution about opening the door for assumptions [embodied in constraints] "to be heralded as geologic evidence". The simple antidote is to formalize model reporting and document the reason for each constraint, as recommended by Flowers et al. (2015), reinforcing the idea that every constraint should have a reason for being there, and the modeler should be able to state what that reason is. If the evidence underlying a constraint is uncertain, then models can be run with and without it, and the effects evaluated – simple!

We agree that this section can be better expressed and we will rewrite this. Constraint boxes are important and necessary and should be based on solid geological evidence. The 'reporting' of reasoning behind a geologic constraint does not universally justify its imposition within a model. A constraint can have a reason for being there but still be incorrect, especially in deep time (see basement nonconformity example mentioned previously). We agree that solutions that violate reliable geological constraints should be excluded. There are always mathematical solutions that can fit data that violate geological constraints. Constraints are very important for nondirected Monte Carlo models such as HeFTy and AFTINV to help speed up convergence by focusing the t–T search in more productive regions of solution space. We wanted to point out that these constraints may not improve results if multikinetic behaviour is unrecognized and data are assumed to belong to a single monokinetic population. In the case of deep time problems, geological evidence can be very fragmentary or non-existent so imposition of constraints must be done carefully, or they may have unintended consequences. We agree that geologic information is important but uncertainty surrounding constraints is not easily dealt with. Models *can* be run under different scenarios and the results compared, but how many papers show and report these types of results?

The discussion features almost 2 pages on (U-Th)/He kinetics (line 514-574), which is really not the subject of the paper, and is not really further informed by any of the modeling work presented here. It is basically editorializing, and most can be omitted. It's unclear what the authors mean by the "distorted" thermal history from using uniform-rmr0 RDAAM means that a non-FT-kinetics He model should be used (line 527-529). An issue with the current alternatives (Gerin et al., 2017; Willett et al., 2017) is that they do not allow for variation of alpha recoil damage annealing kinetics at all. If these kinetics vary, these other models would not be able to capture them, either. More work needed…

More work is certainly needed. We will remove this section and merely point out the simple reason for adding AHe data under the assumption that He diffusion is governed (in some way) by apatite composition within the current $r_{mr0}$ framework. These He data can assist in thermal history recovery under our assumptions, and importantly, *if* composition does cause diffusivity to vary, then He date overdispersion will not always be captured by grain size or eU – which is the working assumption in most published literature. Getting into these minutiae devolves into things like questionable data manipulation practices that were mentioned in the reply to Gallagher. The lack of variation in alpha damage kinetics is probably less of an issue than using FT annealing as a damage proxy for He diffusion (also commonly treated as invariant) when we know it is incorrect and an oversimplification. As a note, within QTQt the activation energy ($\Delta E$) required for He to 'escape' a diffusive trap is a parameter that can be resampled for the Gerin et al. model, so there is at least some acknowledgement of parameter uncertainty that can be explored, which was discussed in their paper.

It's not clear how the authors calculated their rmr0 value for Itambe apatite (line 137); I think it might be that they excluded Si from the "others" category. This should be clarified. It's not clear what they are trying to say in this part, and going back and forth between the two rmr0's from 1999 and 2007 is likely to be confusing unless things are very clearly set out.

Yes, we excluded Si from the 'Others' category since the Carlson 1999 stipulation was that this was intended for substituting cations at the Ca-site (M site) in apatite (general formula $M_{10}(ZO_4)_6X_2$), whereas

Si substitutes at the phosphorus or Z site, therefore we are unsure of the applicability of that assignment for $r_{mr0}$ characterization. We can see that it is confusing after reading reviewers' comments. We will rewrite this to eliminate confusion.

The authors should make it clear that the 0.882 value of rmr0 for end-member OH-apatite (line 185) is an adaptation of fitted rmr0-$\kappa$ values of HS apatite (0.8559, 0.2206) to the simplification that rmr0 + $\kappa$ = 1, providing approximately the same closure temperature. At least, I think that's what they did…

Yes, it is what we did. We will clarify the text, mostly by removing the details of the $r_{mr0}$ discussion from this paper.

---

## Referee Report (RR1)

**McDannell and Issler.. Simulating sedimentary cycles**

**The authors have made substantial changes to the original manuscript and it is now more accessible/readable.**

**They have reduced the length and I think it is now clearer in terms of addressing the role of multi-kinetic (or multi-compositional) data (ages, lengths) in resolving thermal history information. I could make comments on the presentation - those would be primarily stylistic (e.g. there are many Beatles sentences... long and winding) but that is more a personal choice. I admit to not being a big fan of the Beatles in general.**

Some minor, but not necessarily insignificant, comments

L78...not sure you need the word exaggerated here. There is no need to be too apologetic for using near perfect synthetic data..just state it. The paragraph starting on line 98 is a little repetitive on that point.

L56..not clear what is linear here....if eqn 1 has eCl.

L 57 - make it clear in the equation that this is eCl (if it is) rather than the Cl in the original Ketcham et al. 1999) Cl* value.

L193 regardless of feasibility...you mean geological feasibility...rather than data fitting feasibility...clarify that..you could say that a candidate thermal histories that predicts the data adequately (at least in relation to the current thermal history) can be accepted, regardless of its geological feasibility...as you say a few lines below, but it should be here.

L220 - if you generated the synthetic data with QTQt, then the scatter in the ages is from randomly resampling a binomial distribution for Ns, given Ns+Ni...explained in the appendix of Gallagher 1995. As stated ..Varying Ns/Ni.... is not very specific...the resampling process adds a bit of natural noise (i.e. more or less Poissonian).

Also the length data are generated by drawing the desired number of lengths randomly from the predicted distribution.

 (and also note that neither the central ages nor the single grain ages are used for the data fit, as implied in the reply to referees - it is the same binomial distribution approach to give a conditional probabilility for Ns and Ni, given a predicted Ns/Ni.. or rho_s/rho_i).

L 234...the sampling is in a Bayesian framework, but Bayesian sampling is probably not appropriate....you tune the parameters for sampling (strictly we are using proposal functions...that have a standard deviation as a parameter to tune, for example).

L255 - EX is the mean of the posterior distribution, or the weighting being the posterior probability...now that is Bayesian.

L255 examine..better as illustrate ?

L282 - the ability of QTQt to recover....while you are using QTQt, I think the issue is more general...just say... the ability to recover ?

L 382ish- section 5.2 )- any interest in a comment on taking the sub-groups and modelling each one individually to assess what parts of the thermal history are constrained by which subsets of data ?

L430 for information- there is now a published comment and reply on the Green and Duddy discussion.

L 512 - Conclusions....these are pretty clear but perhaps overly positive in the sense that you may be leaving yourselves open to be criticized of promoting over interpretation - as with real data things will not be so nice - but I expect that will be addressed inthe secobd paper.

---

## Editor Decision (ED1)

**AE decision on revised version of**
**Simulating sedimentary burial cycles: I. Investigating the role of apatite fission track annealing kinetics using synthetic data**
**by McDannell and Issler**

Noah McLean

Reviewers Ketcham and Gallagher indicate that this contribution has improved with the authors' extensive revisions. I agree. The resulting manuscript is more focused, clear and readable. I recommend accepting the revised manuscript, with the technical corrections outlined in the second round of referee reports.

One remaining point of contention concerns the scatter in synthetic grain ages created during forward modeling. This topic is discussed in Ketcham's original review but resulted in no revisions by the authors, as discussed on pages 21 and 22 of their response to reviewers. The issue is re-raised at the top of Ketcham's second report, and I agree with his conclusion that the paper would be stronger with more realistic forward-modeled data. However, I understand the authors' choice to simplify the forward-modeled dataset and accentuate the multi-kinetic nature of the synthetic data and its impact on the inverse-modeled thermal histories.

To increase the clarity of the final manuscript, the authors should quantify the modeled scatter in AFT data, which is described around line 220 in the revised manuscript. The forward modeled age dispersion is also the subject of a comment from Gallagher, with reference to the same line number. If the synthetic data were generated with QTQt, then this should be stated and an inline reference to the Gallagher 1995 appendix should be inserted.

The rest of the reviewers' comments require specific, minor technical corrections, which should be addressed by the authors in their final submission. This includes minor technical corrections/wordsmithing on lines 56, 57, 193, 220 (see above), 234, 255, 282, and optionally '382ish' from the Gallagher referee report. Likewise, Ketcham suggests minor technical corrections to lines 33, 34, 84, 98 (see also comment on line 78 from Gallagher), 100, 103, 111, 129, 226, 402, 488, 496, and 512, and optionally address Ketcham's broader comments for lines 424, 457 and 481.